# Efficacy of ultra-short, response-guided sofosbuvir and daclatasvir therapy for hepatitis C in a single-arm mechanistic pilot study

Barnaby Flower[1,2]*, Le Manh Hung[3], Leanne Mccabe[4], M Azim Ansari[5], Chau Le Ngoc[1], Thu Vo Thi[1], Hang Vu Thi Kim[1], Phuong Nguyen Thi Ngoc[1], Le Thanh Phuong[3], Vo Minh Quang[3], Thuan Dang Trong[1], Thao Le Thi[1], Tran Nguyen Bao[1], Cherry Kingsley[2], David Smith[5], Richard M Hoglund[6,7], Joel Tarning[6,7], Evelyne Kestelyn[1,7], Sarah L Pett[4], Rogier van Doorn[7,8], Jennifer Ilo Van Nuil[1,7], Hugo Turner[9], Guy E Thwaites[1,7], Eleanor Barnes[5,7], Motiur Rahman[1,7], Ann Sarah Walker[4,10,11], Jeremy N Day[1,7], Nguyen VV Chau[3], Graham S Cooke[2]

[1]Oxford University Clinical Research Unit, Ho Chi Minh City, Vietnam; [2]Department of Infectious Disease, Imperial College London, London, United Kingdom; [3]Hospital for Tropical Diseases, Ho Chi Minh City, Vietnam; [4]MRC Clinical Trials Unit at UCL, University College London, London, United Kingdom; [5]Peter Medawar Building for Pathogen Research, Nuffield Department of Medicine, University of Oxford, Oxford, United Kingdom; [6]Mahidol Oxford Tropical Medicine Research Unit, Mahidol University, Faculty of Tropical Medicine, Bangkok, Thailand; [7]Centre for Tropical Medicine and Global Health, Nuffield Department of Medicine, Oxford University, Oxford, United Kingdom; [8]Oxford University Clinical Research Unit, Hanoi, Vietnam; [9]MRC Centre for Global Infectious Disease Analysis, School of Public Health, Imperial College London, London, United Kingdom; [10]Nuffield Department of Medicine, University of Oxford, Oxford, United Kingdom; [11]The National Institute for Health Research, Oxford Biomedical Research Centre, University of Oxford, Oxford, United Kingdom

*For correspondence:
bflower@oucru.org

## Abstract

**Background:** World Health Organization has called for research into predictive factors for selecting persons who could be successfully treated with shorter durations of direct-acting antiviral (DAA) therapy for hepatitis C. We evaluated early virological response as a means of shortening treatment and explored host, viral and pharmacokinetic contributors to treatment outcome.

**Methods:** Duration of sofosbuvir and daclatasvir (SOF/DCV) was determined according to day 2 (D2) virologic response for HCV genotype (gt) 1- or 6-infected adults in Vietnam with mild liver disease. Participants received 4- or 8-week treatment according to whether D2 HCV RNA was above or below 500 IU/ml (standard duration is 12 weeks). Primary endpoint was sustained virological response (SVR12). Those failing therapy were retreated with 12 weeks SOF/DCV. Host IFNL4 genotype and viral sequencing was performed at baseline, with repeat viral sequencing if virological rebound was observed. Levels of SOF, its inactive metabolite GS-331007 and DCV were measured on days 0 and 28.

**Results:** Of 52 adults enrolled, 34 received 4 weeks SOF/DCV, 17 got 8 weeks and 1 withdrew. SVR12 was achieved in 21/34 (62%) treated for 4 weeks, and 17/17 (100%) treated for 8 weeks. Overall, 38/51 (75%) were cured with first-line treatment (mean duration 37 days). Despite a high prevalence of putative NS5A-inhibitor resistance-associated substitutions (RASs), all first-line treatment failures cured after retreatment (13/13). We found no evidence treatment failure was associated with host IFNL4 genotype, viral subtype, baseline RAS, SOF or DCV levels.

**Conclusions:** Shortened SOF/DCV therapy, with retreatment if needed, reduces DAA use in patients with mild liver disease, while maintaining high cure rates. D2 virologic response alone does not adequately predict SVR12 with 4-week treatment.

**Funding:** Funded by the Medical Research Council (Grant MR/P025064/1) and The Global Challenges Research 70 Fund (Wellcome Trust Grant 206/296/Z/17/Z).

## Editor's evaluation

Hepatitis C virus (HCV) infection continues to be a global public health problem with over 70 million infected. The current study provides a response to the WHO call for identifying patients with HCV who could be successfully treated with a shorter duration of direct-acting antiviral (DAA) therapy. It provides valuable knowledge to the ongoing research to shorten DAA therapy duration while maintaining high cure rates. Such efforts would impact both treatment access and achieving WHO elimination goals for HCV.

## Introduction

Direct-acting antiviral (DAA) therapy for hepatitis C (HCV) offers high cure rates to those able to adhere to standard durations of treatment. In low- and middle-income countries, where treatment is limited to second-generation NS5A/NS5B-inhibitor combinations, standard treatment is at least 12 weeks. This duration presents a barrier to successful engagement in care for some populations (*Kracht et al., 2019*; *Petersen et al., 2016*), hampering the elimination of HCV as a public health threat. Novel treatment strategies are required for hard-to-reach individuals such as people who inject drugs and those of no fixed abode.

In Vietnam, DAA therapy remains prohibitively expensive for many of those infected. A standard 12-week course of sofosbuvir and daclatasvir (SOF/DCV) was priced at US$2417–2472 in Ho Chi Minh City (HCMC) in 2019 (*Nguyen Thanh et al., 2019*). Despite the government subsidising 50% of drug costs since, the Ministry of Health estimates only 1000 individuals accessed DAA treatment through health insurance in 2019, and 2700 in 2020 (*Ministry of Health V, 2021*).

The World Health Organization has called for research into predictive factors for selecting persons who could be successfully treated with shorter durations of therapy (*World Health Organization, 2018*), which could expand access to treatment and reduce drug costs. Studies evaluating short-course therapy are challenging for infectious diseases where there are significant clinical risks of failure (e.g., TB and sepsis). However, HCV provides a model where treatment failures can be successfully retreated (*Cooke and Pett, 2021*) allowing exploration of mechanisms underlying successful therapy.

Shortened DAA therapy is associated with disappointing rates of cure, such that it could never be recommended routinely. A systematic review and meta-analysis into treatment optimisation for HCV with DAA therapy in individuals with favourable predictors of response, found that pooled sustained virological response (SVR) for regimens of ≤4 weeks duration was 63.1% (95% confidence interval [CI] 39.9–83.7), 6 weeks duration was 81.1% (75.1–86.6) and 8 weeks duration was 94.2% (92.3–95.9) (*Jones et al., 2019*). However improved rates of cure were seen with an increased number of individual-level factors known (or assumed) to be favourable, such as non-genotype 3 infection, lower body mass index (BMI), lower baseline viral load, mild liver disease, absence of prior treatment failure and a rapid virological response to treatment (*Jones et al., 2019*).

Rapid virological response offers a promising means of shortening treatment duration while maintaining high rates of cure. So-called response-guided therapy (RGT), whereby antiviral duration is shortened in individuals who rapidly suppress virus levels in blood after starting treatment, was routinely used in the era of interferon-based therapy, when an undetectable HCV RNA at 4 weeks

**eLife digest** Hepatitis C is a blood-borne virus that causes thousands of deaths from liver cirrhosis and liver cancer each year. Antiviral therapies can cure most cases of infection in 12 weeks. Unfortunately, treatment is expensive, and sticking with the regimen for 12 weeks can be difficult. It may be especially challenging for unhoused people or those who use injection drugs and who have high rates of hepatitis C infection.

Shorter durations of therapy may make it more accessible, especially for high-risk populations. But studies of shorter antiviral treatment durations have yet to produce high enough cure rates. Finding ways to identify patients who would benefit from shorter therapy is a key goal of the World Health Organization.

Potential characteristics that may predict a faster treatment response include low virus levels before initiating treatment, patient genetics, drug resistance mutations in the virus, and higher drug levels in the patient's blood during treatment. For example, previous research showed that a rapid decrease in virus levels in a patient's blood two days after starting antiviral therapy with three drugs predicted patient cures after three weeks of treatment.

To test if high cure rates could be achieved in just four weeks of treatment, Flower et al. enrolled 52 patients with hepatitis C in a study to receive the most widely accessible dual antiviral treatment (sofosbuvir and daclatasvir). Participants received four or eight weeks of treatment, depending on the amount of viral RNA in their blood after two days of treatment. The results indicate that a rapid decrease in virus levels in the blood does not adequately predict cure rates with four weeks of two-drug combination therapy. However, eight weeks may be highly effective, regardless of viral levels early in treatment.

Thirty-four individuals with low virus levels on the second day of treatment received four weeks of therapy, which cured 21 or 62% of them. All seventeen individuals with higher viral levels on day two were cured after eight weeks of treatment. Twelve weeks of retreatment was sufficient to cure the 13 individuals who did not achieve cure with four weeks of therapy. Even patients with drug resistance genes after the first round of therapy responded to a longer second round.

Flower et al. show that patient genetics, virus subtype, drug levels in the patient's blood, and viral drug resistance genes before therapy, were not associated with patient cures after four weeks of treatment. Given that retreatment is safe and effective, larger studies are now needed to determine whether eight weeks of therapy with sofosbuvir and daclatasvir may be enough to cure patients with mild liver disease. More studies are also necessary to identify patients that may benefit from shorter therapy durations. Finding ways to shorten antiviral therapy for hepatitis C could help make treatment more accessible and reduce therapy costs for both individuals and governments.

was used to determine a shorter course of pegylated interferon and ribavirin (*Fried et al., 2011*). Evidence supporting RGT with DAAs at earlier time points is emerging (*Cooke and Pett, 2021*; *Lau et al., 2016*; *Yakoot et al., 2017*), notably using day 2 (D2) viral load to determine treatment duration in genotype 1b infection. In this population, high cure rates were observed with just 3 weeks triple therapy (protease inhibitor [PI], NS5A inhibitor and NS5B inhibitor) (*Lau et al., 2016*). In a UK treatment shortening study, which used 4–8 weeks ombitasvir, paritaprevir, dasabuvir and ritonavir based on baseline viral load, all 10 individuals who became undetectable at D3 of treatment achieved first-line SVR12 regardless of treatment duration(*Cooke and Pett, 2021*). There is currently no data for RGT durations less than 8 weeks with SOF/DCV, which remains the lowest-priced and most widely available treatment option globally (*Clinton Health Access Initiative, 2021*).

Drug resistance in association with particular viral genotypes and subtypes is also known to influence treatment outcome (*Silva Filipe et al., 2017*; *Gupta et al., 2019*) and may predict who can be treated with shortened therapy. Vietnam has a high burden of genotype 6 HCV infection (around 35%) (*Irekeola et al., 2021*), which is rare outside South East Asia and under-represented in clinical trials. Genotype 6 is the most genetically diverse HCV lineage (*Hedskog et al., 2019*), raising concerns about the potential for emergence of resistant variants (*McPhee et al., 2019*).

The human *IFNL4* di-nucleotide polymorphism rs368234815 (ΔG/TT) controls generation of the IFNL4 protein and is also associated with impaired clearance of HCV *Prokunina-Olsson et al., 2013*

and inferior responses to pegylated interferon-alpha/ribavirin therapy (*Franco et al., 2014*) and SOF-based treatment (*Ansari et al., 2017*; *Ansari et al., 2019*). The impact of host *IFNL4* genotype in shortened DAA therapy is not well understood. It is also unknown how serum levels of SOF, its metabolite GS-331007, and DCV might impact treatment success with shortened therapy.

In this prospective single-arm mechanistic study in HCMC, individuals with genotypes 1 and 6 HCV infection and mild liver disease were treated with shortened course SOF/DCV. We tested the hypothesis that high rates of cure can be achieved with short-course DAAs when early on-treatment virological response is used to guide duration of therapy. We also compared host *IFNL4* genetic polymorphism, DAA drug levels, HCV subtypes and previously defined (in vitro) resistance-associated substitutions (RASs), in cures versus treatment failures to better understand the biological mechanisms determining treatment outcome.

## Methods

### Study population

Participants were recruited from the outpatient hepatitis clinic of the Hospital for Tropical Diseases (HTD) in HCMC, between February 2019 and June 2020. Eligible patients were ≥18 years and had chronic infection with HCV genotype 1 or 6 without evidence of liver fibrosis (defined as a FibroScan score≤7.1 kPa, equivalent to F0-F1 disease) (*Nitta et al., 2009*). In addition, participants were required to be HCV-treatment naïve, have a BMI≥18 kg/m$^2$, a creatinine clearance≥60 ml/min, with no evidence of HIV or Hepatitis B coinfection, or solid organ malignancy in the preceding 5 years. Full eligibility criteria are provided in the protocol available at https://doi.org/10.1186/ISRCTN17100273.

Patients referred to the trial were initially enrolled into an observational study which included FibroScan assessment and genotyping. Individuals in this cohort found to be potentially eligible for the trial were invited for further screening. All patients provided written informed consent.

### Study design

All participants were treated with sofosbuvir 400 mg and daclatasvir 60 mg (Pharco Pharmaceuticals, Egypt) administered orally as two separate tablets, once daily. Individuals requiring dose adjustment for any reason were excluded.

Treatment duration was determined using hepatitis C viral load measured 2 days after treatment onset (D2). Participants with viral load <500 IU/ml at D2 (after two doses of SOF/DCV) were treated with 4-week SOF/DCV. Those with HCV RNA≥500 IU/ml received 8 weeks (*Figure 1*). The aforementioned study by *Lau et al., 2016* reported 100% SVR12 following 3-week triple therapy using this threshold. We chose a minimum 4-week duration based on our broader inclusion criteria and the use of dual-class therapy.

To determine viral kinetics on treatment (and on occasion of any failure), HCV viral load was measured at baseline (day 0) and at all subsequent follow-up visits on days 1, 2, 7 and then twice weekly until end-of-treatment (EOT) (*Figure 1*). Visits after EOT were scheduled twice weekly in the first month after completion of treatment, and then at 8 and 12 weeks after EOT.

The primary endpoint was sustained virological response (SVR12) defined as plasma HCV RNA less than the lower limit of quantification (<LLOQ) 12 weeks after the EOT without prior failure. Failure of first-line treatment was carefully defined to incorporate individuals who fully suppressed HCV RNA (<LLOQ) on therapy with late virological rebound, as well as those who never fully suppressed HCV

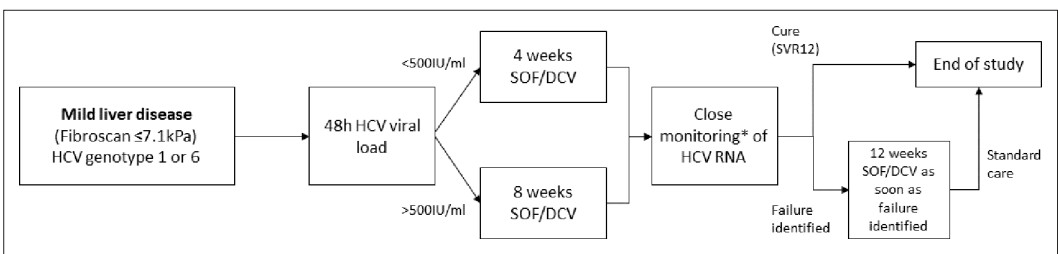

**Figure 1.** Study design. *HCV RNA on days 0, 1, 2, 7, 10, 14, 17, 21, 24, 28, (42, 56), EOT +3, EOT +7, EOT +10, EOT +14, EOT +17, EOT +21, EOT +24, EOT +28s, EOT +56, EOT +84.

viral load. In both cases, two consecutive viral loads >LLOQ, taken at least 1 week apart, were required to confirm failure, with the second >2000 IU/ml. Once failure was confirmed, participants commenced retreatment with standard duration SOF/DCV within 2 weeks (*Figure 1*).

Secondary endpoints were lack of initial virological response (<1 log10 decrease in HCV viral load from baseline), serious adverse events (SAEs), grade 3/4 clinical adverse events (AEs), AEs of any grade leading to change in treatment (SOF, DCV or any other concomitant medication) and adverse reactions (ARs). Severity of all AEs and ARs were graded using the Common Toxicity Criteria for Adverse Events gradings (*National Institute of Health, 2017*).

## Sample size justification

We set a target cure rate of ≥90%, and an unacceptably low cure rate of 70%. Assuming 90% power and one-sided α=0.05, 37 participants were required to exclude the null hypothesis that cure was <90%. Assuming 5% loss to follow-up, and that, based on the study by *Lau et al., 2016*, 65% would suppress viral load <500 IU/ml by day 2 and receive 4 weeks (rather than 8 weeks) of therapy, the final target population was 60 participants, pooling genotypes 1 and 6.

## Study assessments

At each visit, patients were assessed by a study doctor. AEs and ARs were recorded and graded according to a standardised scale (*National Institute of Health, 2017*) and medication adherence and use of healthcare facilities were recorded on case report forms.

HCV RNA was measured in the hospital using the available commercial platform. At start of study (for the first 41 participants enrolled), this was the Abbott Architect (LLOQ=12 IU/ml). This was subsequently replaced with the COBAS AmpliPrep/COBAS TaqMan HCV Quantitative Test, version 2.0 (Roche Molecular Systems, LLOQ=15 IU/ml). Standard laboratory tests—including full blood count, renal function and liver function tests—were performed in the hospital laboratory at baseline, EOT and EOT+12.

## Virus sequencing

At screening, HCV genotype and subtype were determined using NS5B, Core and 5' UTR sequencing, according to the method described by *Le Ngoc et al., 2019*. To evaluate the impact of HCV subtypes and RASs on treatment outcome, whole-genome sequencing (WGS) was additionally performed on all enrolled participants' virus at baseline, and upon virological rebound and at start of retreatment in participants failing therapy. WGS of the HCV viral genome was attained using Illumina MiSeq platform as described previously (*Thomson et al., 2016*; *Smith et al., 2021b*; *Smith et al., 2021a*; *Manso et al., 2020*). The de novo assembly's nucleotide sequences were translated into amino acid and were aligned to H77 HCV reference (GenBank ID: NC_038882.1) and the NS5A and NS5B protein regions were extracted. We only looked for RAS that were present in at least 15% of the reads in the sample and had a read count of greater than 10.

We used the Public Health England (PHE) HCV Resistance Group's definition for RASs (*Bradshaw et al., 2019*). For genotype 1 we looked for RASs defined specifically for genotype 1 as they are well studied. For genotype 6 we looked for all RASs defined across all genotypes, as little work has been done on RASs in genotype 6.

For DCV, we looked for 24R, 28T, 30E/K/T, 31M/V, 32L, 58D and 93C/H/N/R/S/W in genotype 1 infection and additionally looked for 28S, 30R and 31F in genotype 6 infection. For SOF, we looked for 159F, 237G, 282T, 315H/N and 321A/I in genotype 1 infection and additionally looked for 289I in genotype 6 infection (*Ansari et al., 2017*; *Ansari et al., 2019*).

In addition to viral sequencing, we evaluated host genetic polymorphisms within the interferon lambda 4 (*IFNL4*) gene of all participants at baseline. Genotyping of *IFNL4* rs368234815 was performed on host DNA using the TaqMan SNP genotyping assay and primers described previously (*Prokunina-Olsson et al., 2013*) with Type-it Fast SNP Probe PCR Master Mix (Qiagen).

## Pharmacokinetics and pharmacodynamics

To assess pharmacokinetics (PK) and pharmacodynamics (PD), the plasma drug levels of SOF, its inactive metabolite GS-331007, and DCV were measured at baseline, at day 14 and at EOT (day 28 or 56) in all participants. In addition, intensive drug level sampling was conducted in a subset of 40

participants, who were sequentially invited to join an ancillary PK study. In this subgroup, five samples were collected in each participant after the first dose of SOF/DCV and at day 28, according to one of two randomly assigned sampling schedules (A and B). In sampling schedule A, drug levels were measured at 0.5-, 2-, 4-, 6- and 24-hr post-dose; in sampling schedule B, drug levels were measured at 1-, 3-, 5-, 8- and 24-hr post-dose.

Drug quantification was performed using liquid chromatography-tandem mass spectrometer at Mahidol Oxford Tropical Medicine Research Unit, Bangkok. Two separate analytical assays were developed and validated to quantify SOF plus its metabolite GS-331007, and DCV, respectively. Full methodological details of the PK/PD analysis are provided in Appendix 1.

## Statistical analysis

### Primary and secondary outcomes

Analysis was performed under intention-to-treat (the per-protocol analysis, defined as including all participants taking 90–110% of prescribed treatment, was equivalent to the intention-to-treat analysis) with an additional post hoc analysis excluding those who were non-Gt1/6 from WGS. Where possible, proportions and 95% CIs were estimated from the marginal effects after logistic regression. Where no events were recorded and models would not converge, we used binomial exact 97.5% CIs. Absolute HCV VL was analysed using interval regression (incorporating censoring at the LLOQ) adjusting for baseline HCV VL. Differences between baseline means and medians in 4-week cures versus 4-week failures were analysed with unpaired t-tests and Wilcoxon rank-sum tests, respectively; differences in proportions were assessed using chi-squared tests or Fisher's exact tests as appropriate. Analyses were performed using Stata v16.1 (*StataCorp, 2019*).

## Virus genomics

Fisher's exact test was used to test for association between presence and absence of each RAS and treatment outcome. To test for association between outcome and number of RAS, we used logistic regression.

## Pharmacokinetics and pharmacodynamics

Intensive drug levels of SOF, its metabolite GS-331007, and DCV from the subset of 40 patients at days 0 and 28, together with any EOT samples at day 28, were analysed using non-compartmental analysis in PKanalix version 2020R1 (*Lixoft, 2022*). Two separate analyses were performed to characterise the pharmacokinetic properties of the study drugs.

In the first, naïve pooled analyses were performed separately on data from days 0 and 28 (not including EOT samples) to derive median pharmacokinetic parameters at each day. In these analyses, the median concentration at each protocol time was calculated. Individual concentration measurements below the LLOQ were set to LLOQ/2 when calculating the median values. It was assumed that the participants had no drug concentrations at time 0.

In the second analysis, data from days 0 and 28 were pooled for each individual. This resulted in a full pharmacokinetic profile for each subject, which was analysed with a non-compartmental approach. The mean value of drug concentrations was used if patients had two or more samples taken at the same time point. These derived individual drug exposures were used to evaluate the relationship between drug exposure and therapeutic outcome. It was assumed that the participants had no drug concentrations at time 0. In this analysis, the first measurement below LLOQ in a series of LLOQ samples was imputed as LLOQ/2 and the later measurements were ignored. In both approaches, SOF samples taken at ≥24 hr post-dose were excluded. SOF is a prodrug and has a very short half-life of less than 1 hr, which make concentrations at 24 hr after dose extremely unlikely (*de Kanter et al., 2014*). In addition, outcome variables and the relationship between outcome variables and drug exposure were evaluated. Additional detail of the PK/PD analysis is provided in Appendix 1.

## Ethical approval

The trial was approved by the research ethics committees of The Hospital for Tropical Diseases (*Hospital for Tropical Diseases Ethics Commitee, 2021*) (ref: CS/BND/18/25), *Vietnam Ministry of Health, 2022* (ref: 6172/QĐ-BYTtnam MoH), Imperial College London (*Imperial College Research Ethics Committee, 2018*) (ref: 17IC4238), and Oxford University Tropical Research Ethics

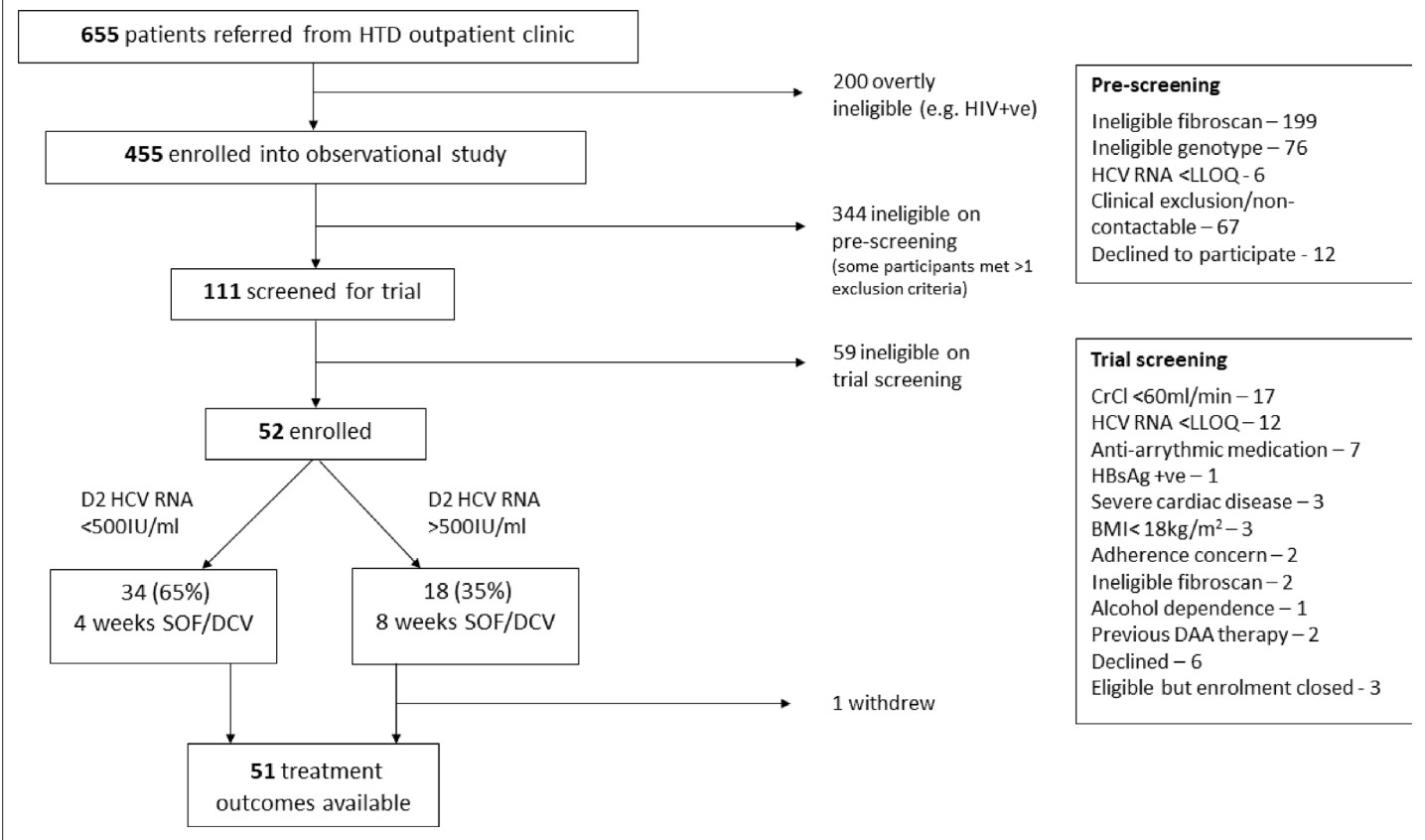

**Figure 2.** Screening and enrolment.

Committee (*Oxford Tropical Research Ethics Committee, 2018*) (ref: 43-17). The study's conduct and reporting is fully compliant with the World Medical Association's Declaration of Helsinki on Ethical Principles for Medical Research Involving Human Subjects (*World Medical Association, 2022*). The trial was registered at ISRCTN, registration number is ISRCTN17100273 (*ISRCTN registry, 2018*).

## Results

### Baseline characteristics

Of 455 patients screened, 52 were enrolled and 1 subsequently withdrew (*Figure 2*). Most exclusions were on account of either a FibroScan score of >7.1 kPa (with cirrhotic patients enrolled into a parallel study; *Flower et al., 2021*), or ineligible genotype.

22/51 were initially identified as genotype 1 infection and 30 as genotype 6. With the benefit of WGS data, it was confirmed that 22 (43%) had genotype 1 infection, 27 (53%) had genotype 6, 1 had genotype 2 and another had genotype 4 infection. The latter two individuals were included in the intention-to-treat analysis but excluded from a post hoc analysis of G1 and G6 infections only.

Recruitment was completed short of the initial target of 60 due to severe COVID-19-related restrictions in Vietnam from February 2020. These included clinic closures, travel restrictions and repurposing of the HTD as a COVID-19 treatment centre.

Baseline and clinical characteristics are described in *Table 1*. One participant, a male with genotype 1b infection who was cured with 4-week therapy, had an HCV viral load of 618 IU/ml on day 0 which may have been consistent with spontaneously resolving acute infection, but could equally reflect fluctuating viraemia. Baseline viral load was >10,000 IU/ml in all other participants, who were all assumed to have chronic infection.

**Table 1.** Baseline characteristics.

| | N/ median | %/range |
|---|---|---|
| Total participants | 52 | |
| Age in years | 49.5 | (25.0, 67.0) |
| Female | 29 | (56%) |
| Body-mass index in kg/m² | 23.3 | (18.7, 30.6) |
| Genotype 1 | 22 | (43%) |
| 1a | 11 | |
| 1b | 12 (1 withdrew) | |
| Genotype 6 | 27 | (53%) |
| 6a | 12 | |
| 6e | 10 | |
| 6h | 2 | |
| 6l | 2 | |
| 6u | 1 | |
| Genotype 2(m) | 1 | |
| Genotype 4(k) | 1 | |
| Baseline HCV viral load in IU/ml | 1,932,775 | (618, 11,200,000) |
| HCV viral load – log10 IU/ml (range) | 6.3 | (2.8, 7.0) |
| **Past medical history:** | | |
| Illicit drug use | 4 | (8%) |
| Alcohol dependence (historic; current excluded) | 4 | (8%) |
| Diabetes | 2 | (4%) |
| Hypertension | 7 | (13%) |
| Ischaemic heart disease | 1 | (2%) |
| Tuberculosis | 2 | (4%) |
| Current smoker | 18 | (35%) |
| Previous spontaneous clearance of HCV with re-infection | 2 | (4%) |

## Treatment duration, adherence and efficacy outcomes

By day 2, 34 participants (65%) had HCV viral load below the threshold of 500 IU/ml (*Figure 2*; *Table 2*), so received 4-week treatment. Eighteen participants were above the threshold at this time point, of which 1 withdrew after 9 days of treatment, meaning 17 completed 8-week therapy. Adherence was good, with 96% completing the full prescribed course of SOF/DCV (as assessed by self-reporting and physician pill count). Eighteen (35%) participants missed at least one visit because of COVID-19-related restrictions. Of the 51 participants with outcome data, 38 (75% [95% CI (63, 86)]) achieved SVR12 while 13 failed therapy and required retreatment. All treatment failures occurred in individuals who received 4-week therapy, translating to an SVR12 of 62% (21/34; 95% CI (44, 78)) in rapid responders who received 4-week therapy, and 100% (17/17; 97.5% CI (80, 100)) in slower responders who received 8-week SOF/DCV (*Figure 3*; *Table 2*).

Of the 13 participants who underwent retreatment, 100% were cured. The mean first-line SOF/DCV treatment duration was 37 days (standard deviation, SD 13.7), with a first-line cure rate of 75%. The mean (SD) total SOF/DCV duration (i.e., including 12-week retreatment where required), was 58 (34.2) days per patient, with a 100% cure rate. There was no evidence of differences in age, gender, BMI, IFNL4 genotype, transaminases or baseline HCV viral load between patients who achieved cure with 4-week treatment versus those who experienced treatment failure with 4-week treatment (*Table 3*).

## Viral kinetics and timing of treatment failure

All participants had an initial virological response (i.e., ≥1 log10 decrease in HCV viral load from baseline) (*Appendix 1—figures 1 and 2*). There was no evidence of association between time to complete virological suppression (HCV RNA<LLOQ) and treatment outcome (*Table 3*; *Appendix 1—figures 2 and 4*). In an exploratory analysis, we estimated first-line cure rates based on suppression below the LLOQ at other time points, which could be used for RGT. At day 7, 9/21 cures and 1/12 treatment failures (one missed visit) had HCV RNA<LLOQ (p=0.054; *Table 3*), translating to 90% sensitivity (95% CI [56, 100]) for predicting cure with 4-week treatment. However, by day 10, 9/21 cures and 9/13 failures had HCV RNA<LLOQ (p=1.00), making a rapid virological response 50% [26, 74] sensitive in predicting cure with 4-week treatment. HCV RNA kinetics in all participants treated with 4-week SOF/DCV are shown in *Appendix 1—figure 2*, with cures (blue lines) distinguished from those experiencing treatment failure (red lines). Even though the numbers

**Table 2.** Treatment outcome.

| | N/median | %/range |
|---|---|---|
| Detectable HCV viral load (HCV VL) at day 2 | 50 | 96% |
| Abbott | 39/41 | 95% |
| COBAS | 11/11 | 100% |
| Median (IQR) HCV VL at day 2 in IU/ml | 269 | (104, 690) |
| Abbott | 217 | (101, 690) |
| COBAS | 459 | (209, 832) |
| Below threshold—for 4-week therapy | 34 | (65%) |
| Abbott | 31 | (66%) |
| COBAS | 3 | (60%) |
| Above threshold—for 8-week therapy | 18 | (35%) |
| Abbott | 16 | (34%) |
| COBAS | 2 | (40%) |
| Mean (SD) duration of first-line therapy received in days | 37 | (13.7) |
| Mean (SD) duration of all therapy received in days | 58 | (34.2) |
| Median weeks from enrolment to last visit (range) | 20 | (1, 42) |
| **Primary outcome** | | |
| Outcome available | 51 | |
| SVR12 by intention-to-treat analysis and per protocol analysis | 38 | (75% [95% CI 63, 86]) |
| SVR12 by sensitivity analysis (i) [missing results = failure] | 38 | (73% [95% CI 61, 85]) |
| SVR12 by post hoc analysis (ii) [G1 and G6 only] | 37 | (76% [95% CI 63, 88]) |
| **Secondary endpoints** | | |
| Lack of initial virological response | 0 | (0% [97.5% CI 0, 7]) |
| Serious adverse events | 0 | (0% [97.5% CI 0, 7]) |
| Grade 3/4 clinical adverse events | 0 | (0% [97.5% CI 0, 7]) |
| Non-serious adverse reactions | 18 | (35% [95% CI 22, 48]) |
| Adverse events or reactions leading to change in study medication | 0 | (0% [97.5% CI 0, 7]) |

Where not labelled, data presented as n (%; 97.5% confidence interval).

are small, this helps illustrate that early on-treatment response alone may be of limited value in determining cure with ultra-short therapy.

Since the two HCV assays used in our study have previously been shown to yield different HCV RNA results in the same individuals on therapy (*Maasoumy et al., 2016*), we conducted additional analyses of viral kinetics stratified by platform. We found no evidence of a difference between platforms in terms of proportion of participants with undetectable viral load at different time points (*Table 2*, *Table 3*), or in terms of first phase (days 0 to 2) or second phase (day 2 to first HCV RNA<LLOQ) viral decline on treatment (*Appendix 1—figure 3*). However, numbers were small meaning we may have lacked power to detect effects.

All treatment failures occurred during follow-up after EOT. Despite intensive twice weekly sampling from EOT to EOT +28d, the earliest virologic rebound occurred 3 weeks after completion of therapy (*Appendix 1—figure 5*). Pseudo-anonymised raw viral load data from this study is available in *Source data 1*.

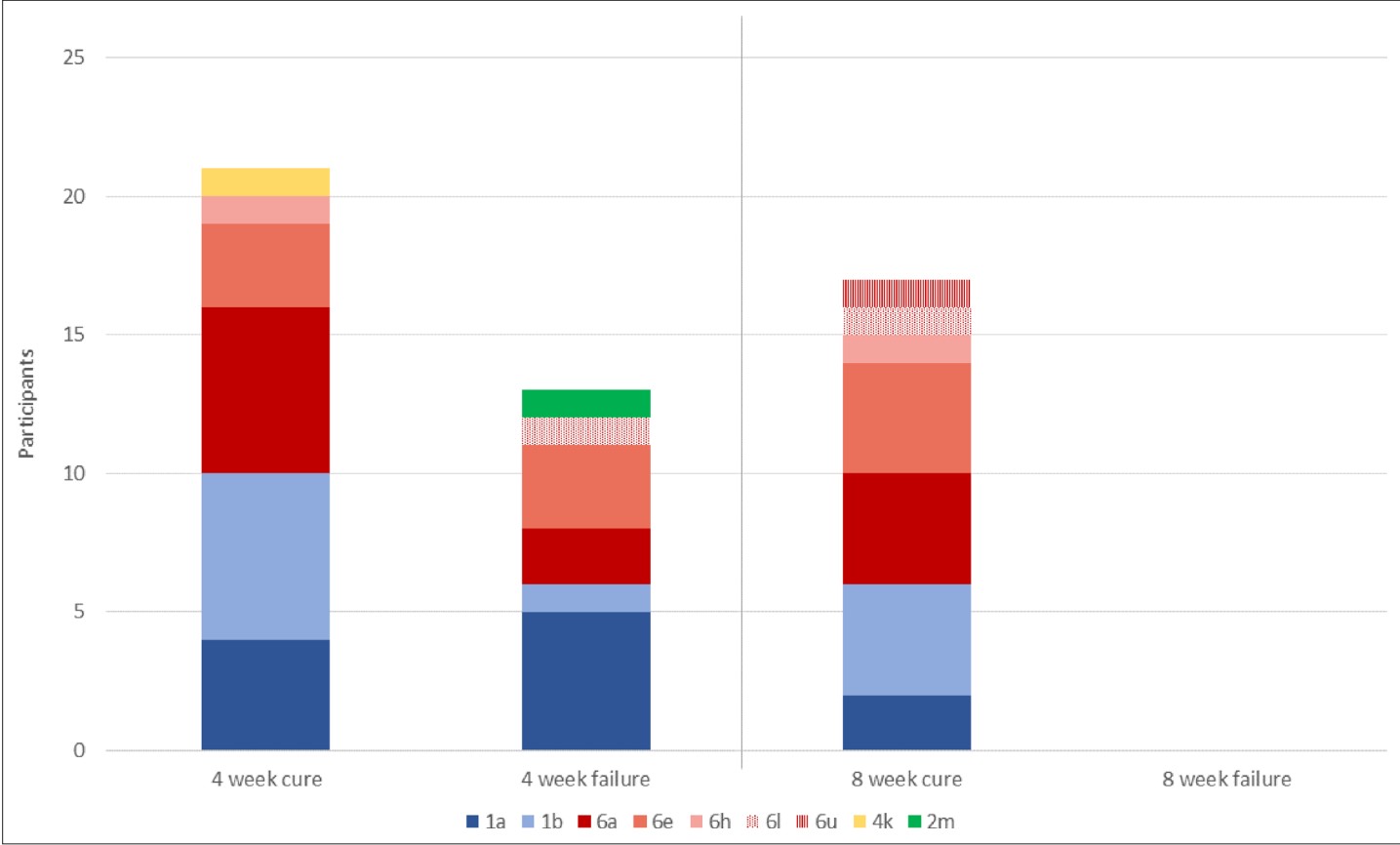

**Figure 3.** Primary outcome, with HCV subtypes (n=51). All 13 individuals who experience treatment failure with 4-week SOF/DCV were cured with 12-week SOF/DCV retreatment.

## Viral genomics at baseline

WGS was attempted on all participants' virus at baseline, but consensus sequences could not be assembled in two individuals (who had low baseline viral load and were both cured with first-line therapy). This left 50 patients with baseline sequences, of which 49 had outcome data.

We found nine discrepancies between lab genotyping and sequencing-based genotyping. Five of these differences were at the level of subtypes for genotype 6 samples, highlighting difficulties inherent in classifying this rare and genetically diverse lineage using an amplicon approach for genotyping (lab genotyping). Two samples were called 6a/e using lab genotyping and WGS classified them as 6e. One sample was classified as 6e on lab genotyping, but WGS showed that it was a genotype 2m sample. WGS revealed another patient to have mixed infection with genotype 1a and genotype 6a; this had been classified by laboratory genotyping as a genotype 6a mono-infection. The individual with mixed infection received 4-week SOF/DCV but cure was not achieved, with relapse of the genotype 1a infection. They subsequently responded to 12-week retreatment.

We found no evidence of differences between genotypes or subtypes with regard to rates of treatment failure. Among genotype 1-infected individuals, 1/7 subtype 1b infections experienced treatment failure with 4-week therapy compared with 4/8 subtype 1a infections (including the mixed infection case) (p=0.15). Among genotype 6-infected individuals, 1/8 subtype 6a infections were not cured with 4-week SOF/DCV compared with 3/6 subtype 6e (p=0.58), 0/1 subtype 6h and 1/1 subtype 6l.

At baseline, the 159F SOF RAS was identified in one patient, and the 237G putative SOF RAS was identified in six patients (*Appendix 1—figures 6 and 7*). The DCV RAS 24R, 30R, 31M, 93H and 93S were detected at baseline (*Appendix 1—figures 8 and 9*).

In the assessment of SOF RAS (*Appendix 1—figure 6*), one patient who had 159F at baseline failed treatment, although this was a minority variant making up 20% of the sequencing reads (*Figure 4*).

**Table 3.** Comparison of baseline factors, drugs levels and virological response in individuals failed to achieve SVR12 with 4-week therapy versus those who cured with 4- or 8-week therapy.

| | 4-week cures (n=21) | 4-week failures (n=13) | p | 8-week cures (n=17) |
|---|---|---|---|---|
| **Host factors** | | | | |
| Male (%) | 62% | 38% | 0.18 | 29% |
| Mean age | 45 | 48 | 0.23 | 55 |
| Mean BMI | 23 | 23 | 0.40 | 24 |
| Median ALT | 54 | 36 | 0.10 | 31 |
| Median AST | 34 | 28 | 0.44 | 33 |
| IFNL4 delG/TT and TT/TT genotypes (rs368234815) | 71% | 58% | 0.47 | 69% |
| **Virus factors** | | | | |
| Median D0 HCV VL | 916,000 | 2,139,258 | 0.20 | 4,982,889 |
| Abbott | 960,913 | 1,972,841 | 0.47 | 4,625,118 |
| COBAS | 916,000 | 5,260,000 | 0.40 | 4,605,000 |
| D2 VL<LLOQ | 2/21 (10%) | 0/13 (0%) | 0.51 | 0% |
| Abbott | 2/18 (11%) | 0/10 (0%) | 0.41 | 0/13 (0%) |
| COBAS | 0/3 (0%) | 0/3 (0%) | – | 0/5 (0%) |
| D7 VL<LLOQ | 9/21 (43%) | 1/12 (8%)* | 0.054 | 0% |
| Abbott | 8/18 (44%) | 1/9 (11%) | 0.09 | 0/13 (0%) |
| COBAS | 1/3 (33%) | 0/3 (0%) | 1.00 | 0/5 (0%) |
| D10 VL<LLOQ | 9/21 (43%) | 9/13 (69%) | 0.17 | 6% |
| Abbott | 8/17 (47%) | 8/10 (80%) | 0.12 | 1/10 (10%) |
| COBAS | 1/4 (25%) | 1/3 (33%) | 1.00 | 0/6 (0%) |
| D14 VL<LLOQ | 14/21 (68%) | 9/13 (69%) | 1.00 | 18% |
| Abbott | 11/16 (69%) | 6/9 (67%) | 1.00 | 1/11 (18%) |
| COBAS | 2/4 (50%) | 3/4 (75%) | 1.00 | 1/6 (17%) |
| HCV genotype 1 | 10/21 (48%) | 6/13 (46%) | 1.00 (vs Gt 6) | 6/17 (35%) |
| 1a | 4/21 (19%) | 5/13 (38%) | 0.15 (vs 1b) | 2/17 (12%) |
| 1b | 6/21 (24%) | 1/13 (8%) | | 4/17 (24%) |
| HCV genotype 6 | 10/21 (48%) | 6/13 (46%) | | 11/17 (65%) |
| 6a | 6/21 (29%) | 2/13 (15%) | 0.58 (vs. 6e) | 4/17 (24%) |
| 6e† | 3/21 (14%) | 3/13 (23%) | | 4/17 (24%) |
| **Resistance-associated substitutions** | | | | |
| Median (range) SOF-RAS | 0 (0–1) | 0 (0–2) | 0.76 | 0 (0–1) |
| Median (range) DCV-RAS | 2 (0–2) | 1 (0–2) | 0.17 | 2 (0–4) |
| Median (range) SOF- & DCV-RAS combined | 2 (0–3) | 2 (1–2) | 0.12 | 2 (0–4) |
| Drug exposure (n=37) § | n=15 | n=8 | | n=14 |

*Table 3 continued on next page*

*Table 3 continued*

|  | 4-week cures (n=21) | 4-week failures (n=13) | p | 8-week cures (n=17) |
|---|---|---|---|---|
| Median AUC$_{last,}$ SOF (h×ng/ml) $^\ddagger$ | 2360 (1120–4550) | 2220 (937–3910) | 0.975 | 2120 (1430–2610) |
| Median AUC$_{last}$ GS-331007 (h×ng/ml) $^\ddagger$ | 11,700 (8420–14,100) | 15,100 (9240–19,700) | 0.023 | 14,000 (10,200–17,400) |
| Median AUC$_{last,}$ DCV (h×ng/mL) $^\ddagger$ | 13,000 (6800–22,300) | 13,200 (6630–27,000) | 0.728 | 14,200 (9210–17,000) |

Results presented as median (5th–95th percentile).
*n=12, no HCV VL data for one participant's day 7 visit.
$^\dagger$h, l and u subtypes excluded from the table/analysis due to small numbers (≤2).
$^\ddagger$AUC$_{last}$ is the total exposure to the last time point (8 hr for SOF and 24 hr for GS-331007 and DCV).
$^\S$Complete d0 and d28 data only available for 37 participants.

237G was identified as a majority variant in two individuals where treatment failed but was also seen in four individuals who were cured (three received 4-week treatment; *Appendix 1—figure 6*).

The most prevalent DCV RAS was 31M, present in nine participants where treatment failed after 4-week first-line therapy (*Figure 4*; *Appendix 1—figures 8 and 9*). However, 31M was also found in 13 individuals cured with 4-week treatment, and 13 cured with 8 weeks. The next most prevalent RAS was 30R, present at baseline in three patients who had treatment failure, in five individuals cured with 4-week treatment and in four patients cured with 8-week treatment. 30R RAS was present in 11/12 6a genomes and 1/1 2m genomes but was absent in other subtypes. 31M RAS was present in 10/11 1a genomes and 12/12 6a genomes and was also found in other subtypes (*Appendix 1—figures 8 and 9*). Additionally, almost all of the subtype 6a samples carried both 30R and 31M RASs while other subtypes did not carry this combination (apart from the 2m sample).

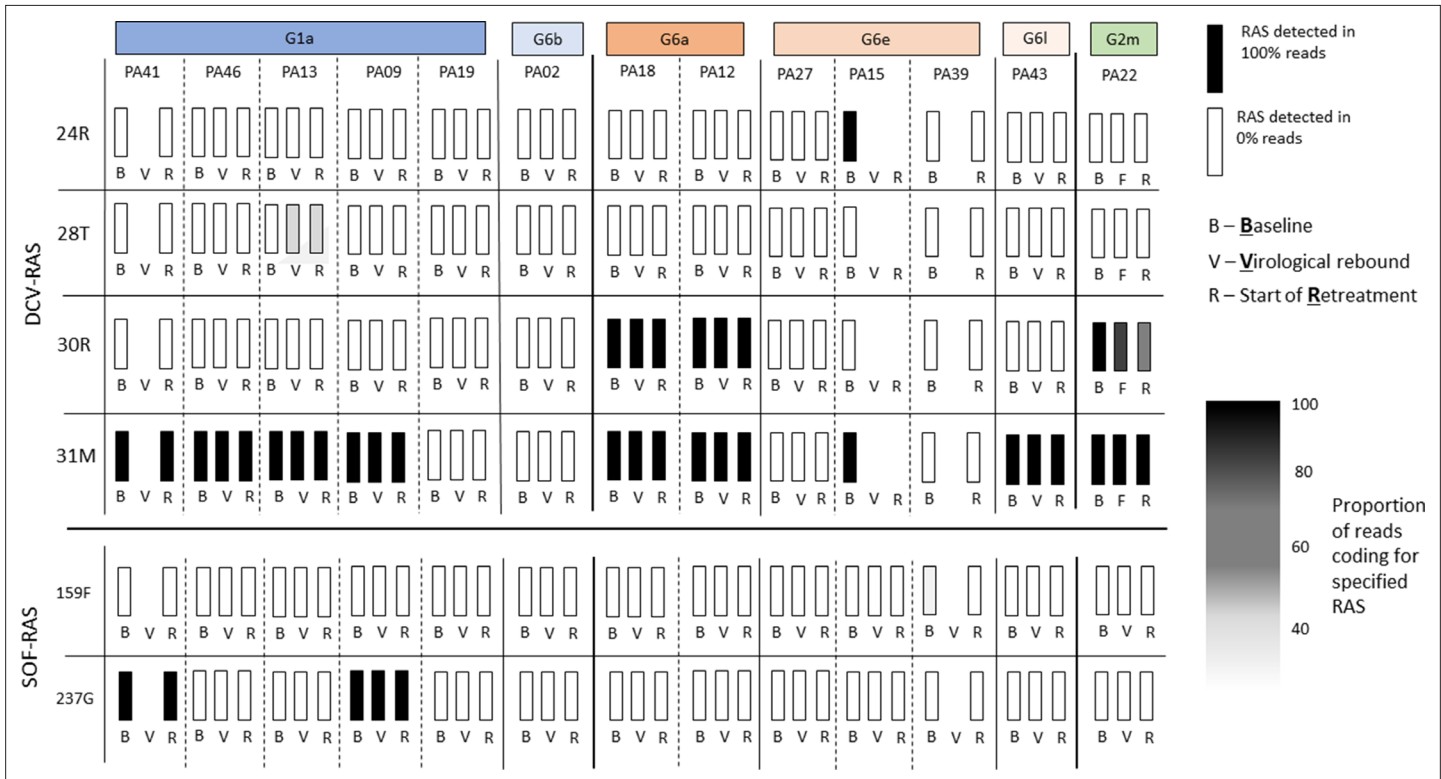

**Figure 4.** Sofosbuvir RAS and Daclatasvir RAS at baseline, treatment failure, and at start of retreatment in all participants who failed first-line treatment.

### Viral genomics in participants failing first-line therapy

Among 13 individuals who experienced treatment failure, we compared the emerging viral genome with baseline virus (*Figure 4*). Full genome sequences could not be assembled for three participants at time of virological relapse; however, we were able to generate whole viral genomes using samples from the start of retreatment for two of these individuals. No new genomes were identified at treatment failure (ruling out any new infections). No new SOF RAS were identified on virologic rebound. DCV 28T RAS (not present at baseline) was identified in one participant failing therapy (*Figure 4*; *Appendix 1—figure 11*) as a minority variant at time of virological rebound and start of retreatment (at 30% and 25% of reads, respectively). Given 100% of retreated individuals achieved SVR12 with standard duration of therapy, we found no evidence to suggest this emerging RAS was clinically significant. There was no evidence of differences in the number of combined SOF- and DCV-RAS at baseline in those who failed 4-week therapy (median 2, range 0–3) versus those who cured with 4 weeks (median 2, range 1–2) (p=0.12), or in those with a slower initial virological response, who received 8 weeks (median 2, range 0–4).

### Pharmacokinetics and pharmacodynamics

Pharmacokinetic parameters derived from the naïve pooled analysis (based on 40 patients on day 0 and 37 patients on day 28) are presented in *Appendix 1—table 1*. Exposure after the individual analysis as well as outcome measurements are presented in *Appendix 1—table 2*. In the individual analysis and the linear regression between outcome measurements and drug exposure, three patients were excluded as they did not have dense samples collected at day 28 (n=37). In the analysis of outcome variables, data from all 40 patients were used. No significant relationship between outcome variables and drug exposure was found using linear regression (*Appendix 1—table 3*).

In the subset of 37 patients who underwent dense PK analysis at days 0 and 28, 23 patients received 4-week SOF/DCV and 14 patients received 8-week therapy. There was no significant difference between total drug exposure ($AUC_{last}$) for SOF and DCV in 4-week cures (n=15) versus 4-week failures (n=8) (*Table 3*). However, GS-331007 exposures were significantly higher in the patient group with treatment failures (p=0.023).

### Safety

SOF/DCV was well-tolerated, and no participants discontinued treatment due to drug side effects. Eighteen participants (35%; 95% CI 22%, 48%) reported at least one non-serious AR. The most common of these were insomnia, gastritis and dizziness, which are all consistent with undesirable effects described in the summary of product characteristics of SOF/DCV (*EMA, 2014*). There were no SAEs or grade 3 or 4 AEs.

## Discussion

In this mechanistic study in individuals with genotype 1 or 6 HCV infection and mild liver disease, treated with 4- or 8-week SOF/DCV depending on HCV viral load 2 days after starting treatment, first-line cure rate was 75% [95% CI (63, 86)], with a mean of 37 days treatment. This saved 47 days of DAA therapy per participant compared with a standard 12-week course, but cure rate fell below our target of ≥90%. For the secondary endpoint—SVR12 after combined first-line therapy or retreatment—cure was 100%, with mean treatment duration of 58 days, saving 26 days DAAs per participant.

### Effect of shortening therapy

Inferior rates of cure are well described when DAA therapy is shortened below 8 weeks without use of early on-treatment virological response, falling below 50% with ≤4 weeks therapy without stratification (*Jones et al., 2019*; *Emmanuel et al., 2017*; *Cooke and Pett, 2021*; *Sulkowski et al., 2017*). A few small studies have reported high rates of cure with shortened therapy based early virological response (*Lau et al., 2016*; *Etzion et al., 2020*; *El-Shabrawi et al., 2018*; ). The only previous RGT study to use less than 6-week treatment, by Lau et al., found a cure rate of 100% with just 3-week DAA therapy in 18 individuals whose HCV viral load was suppressed below 500 IU/ml after 2 days of therapy. This was the same threshold and time point used in our study. One important difference was in the treatment regimen, which included a protease inhibitor (simeprevir or asunaprevir). Although

NS5A- (DCV) and NS5B- (SOF) inhibitors rapidly eliminate HCV from the blood, second-phase decline in viral load appears to be enhanced by addition of a protease inhibitor (*Perelson and Guedj, 2015*). This may be crucial in sustaining high rates of cure with shortened therapy. Viral kinetics in our participants were broadly similar to those observed in patients treated with DCV-containing regimens in the study by Lau et al., with a rapid first-phase viral decline leading to an approximate 4 log10 IU/ml decline in HCV RNA in the first 48 hr of treatment. However, a detailed comparison of viral kinetics is limited by differences in sampling schedule, baseline viral loads and the HCV PCR platforms used. Another key difference relates to infecting genotypes—all participants in the Lau study had genotype 1b infection, compared with just 23% (n=12) in ours. Genotype 1b is associated with favourable outcomes with some DAAs compared with other genotypes (*Ferenci et al., 2014*; *Kohli et al., 2015*; ). Although real-world 1b outcomes with standard duration SOF/DCV appear similar to other non-3 genotypes (*Charatcharoenwitthaya et al., 2020*; ), subtype may be more important when treatment is shortened.

## Role for RGT with SOF/DCV

Cure rates with this strategy are too low for it to be recommended routinely. With standard duration therapy, SVR12 is known not to be impacted by time to first undetectable HCV RNA (*Kowdley et al., 2016*) or by the presence of detectable virus at the EOT (*Pal et al., 2020*). This also appears to be true of shortened treatment: in one individual who experienced treatment failure, HCV viral load was already <LLOQ by day 7; in five of the 4-week cures, HCV VL was only suppressed to <LLOQ virus for the first time at EOT (*Appendix 1—figure 4*). Comparison of 4-week cures and 4-week treatment failures indicates that an HCV RNA<LLOQ by day 7 may be a useful discriminator of 4-week treatment outcome (p=0.054). However, day 10, HCV RNA<LLOQ was not predictive of response to shortened treatment. Day 7 viral load thresholds for shortening DAA therapy are currently being evaluated as part of a large ongoing randomised controlled trial in Vietnam (*McCabe et al., 2020*; ).

## A case for 8-week SOF/DCV therapy

Given the high rates of cure observed with 8-week SOF/DCV in participants with a slow initial virological response (17/17), there is a case for reducing SOF/DCV therapy from 12 to 8 weeks in individuals with mild liver disease. Prior evidence for caution regarding 8-week SOF/DCV comes predominantly from a small 2015 study in HIV-coinfected individuals (*Wyles et al., 2015*), in which 7/10 treatment failures in the 8-week arm received half-dose daclatasvir (30 mg) because participants were taking concomitant darunavir–ritonavir. This dose adjustment was subsequently deemed unnecessary once drug-interaction data emerged, such that this study is likely to underestimate the efficacy of 8-week SOF/DCV. More recent studies corroborate our finding of >90% cure with 8 weeks NS5A/NS5A inhibitor combination (*Yakoot et al., 2017*; *El-Shabrawi et al., 2018*; *Boyle et al., 2020*). Larger trials are warranted to evaluate 8-week SOF/DCV therapy for patients with mild liver disease (irrespective of speed of virological response). This could save significant costs, particularly in countries where pricing is determined per pill rather than per treatment course, such as Vietnam, and the USA (*Emmanuel et al., 2017*; *Clinton Health Access Initiative, 2021*).

## Impact of RASs and retreatment concerns

To our knowledge, this study is the largest assessment of G6 RAS in vivo with SOF/DCV therapy. We hypothesised that a high number of putative RAS at baseline may be associated with higher rates of failure with shortened treatment. However, we found no evidence that number or type of SOF- or DCV-RAS was different at baseline in 4-week cures compared with 4-week treatment failures (*Table 3*, *Appendix 1—figures 6 and 8*), although numbers were small. Additionally, the excellent retreatment outcomes observed (13/13) are reassuring, particularly for low-resource settings where protease inhibitor-based retreatment options are limited. Only one novel RAS was detected after first-line treatment failure, and the individual concerned achieved SVR with standard duration retreatment, suggesting this was not clinically relevant.

## Impact of drug levels

This was the first assessment of the impact of DAA drug levels on efficacy of shortened therapy. The inactive SOF metabolite GS-331007 is the main circulating metabolite of SOF prior to undergoing

renal excretion, and it is frequently used to describe SOF's pharmacokinetics (*Smolders et al., 2019*). We hypothesised that accumulation and slow elimination of GS-331007 and DCV in vivo might protect against the re-emergence of HCV viraemia. However, we found no evidence of a difference in $AUC_{last}$ between 4-week cures and 4-week treatment failures for SOF or DCV. Total exposure to GS-331007 was higher in treatment failures (15,100 [9240–19,700] vs. 11,700 [8420–14,100] [p=0.023]). This was a surprising result, given that SOF and GS-331007 AUCs are near dose proportional over the dose range of 200–1200 mg (*Smolders et al., 2019*), and higher day 10 concentrations of GS-331007 have been associated with improved rates of cure with SOF/ribavirin treatment (*Ahmed et al., 2019*). Further PK studies are warranted to better understand if SOF metabolism impacts treatment outcomes.

## Limitations

Our study has important limitations. First, it was powered to determine the overall cure rate with 4- and 8-week treatment, rather than outcomes with each duration. It is possible that we would have seen patients failing 8-week therapy with a larger sample, and our cure estimates may therefore be imprecise. Second, the participating cohort did not include any individuals with HIV, hepatitis B co-infection or renal impairment and only four participants reported a history of injecting drug use, of which none were currently injecting. These populations are known to have an altered immunological response and constitute an important part of the HCV epidemic. Third, in order to identify the timing of failure, the protocol required a visit schedule with many more visits than is standard of care, which many patients would not be able to follow. Consequently, adherence was very high, which may not reflect real-world practice.

Another potential limitation relates to our use of two different HCV RNA platforms which have previously been shown to give discrepant results in the same individuals (*Dahari et al., 2016*). The Abbott Architect has a lower LLOQ than the COBAS AmpliPrep/COBAS TaqMan and may detect HCV RNA for longer on therapy than the COBAS (*Maasoumy et al., 2016*), though we found no evidence of difference in viral decline by platform. With regard to the PK analysis our non-compartmental analysis of drug levels may not adequately account for drug accumulation of sofosbuvir's metabolite GS-331007 and DCV between days 0 and 28, which was observed (see Appendix 1 for more detail).

In summary, our findings indicate that shortened SOF/DCV therapy cures a significant proportion of patients with mild liver disease without compromising retreatment with the same drug combination in those who fail first-line therapy. This study adds to a growing case for shortening SOF/DCV therapy in individuals with mild liver disease from 12 to 8 weeks, offering retreatment with 12-week SOF/DCV when required. There was no evidence that relatively high numbers of putative RASs at baseline were associated with treatment outcomes, suggesting routine sequencing at baseline or prior to retreatment remains unnecessary. We also found no evidence that drug levels affect virological response or influence treatment outcome. Further work is required to understand which factors predict cure with ultra-short DAA treatment.

## Acknowledgements

This work was supported by the Medical Research Council (Grant MR/P025064/1), The Global Challenges Research Fund (Wellcome Trust Grant 206/296/Z/17/Z). GC is supported in part by the NIHR Biomedical Research Centre of Imperial College NHS Trust and an NIHR Professorship. BF received a travel grant to attend AASLD conference in Washington DC from Gilead Sciences in 2017. HCT acknowledges funding from the MRC Centre for Global Infectious Disease Analysis (reference MR/R015600/1), jointly funded by the UK Medical Research Council (MRC) and the UK Foreign, Commonwealth & Development Office (FCDO), under the MRC/FCDO Concordat agreement and is also part of the EDCTP2 programme supported by the European Union. EB acknowledges support from Oxford NIHR Biomedical Research Centre. MAA is supported by a Sir Henry Dale Fellowship jointly funded by the Royal Society and Wellcome Trust (220171/Z/20/Z). ASW and EB are NIHR Senior Investigators. LM, SLP and ASW are supported by core support from the Medical Research Council UK to the MRC Clinical Trials Unit [MC_UU_00004/03]. JT and RH are supported by the Wellcome Trust (Grant 220211). The views expressed are those of the author(s) and not necessarily those of the NIHR or the Department of Health and Social Care.

The authors would like to thank the patients of the HTD who volunteered to participate in the trial, our hard-working nurses An Nguyen Thi Chau, Tan Dinh Thi, Nga Tran Thi Tuyet, and Phuc Le Thi, as well as members of the data monitoring committee: Hoa Pham Le, Timothy Peto and John Dillon.

## Additional information

### Competing interests

Leanne Mccabe: has received a grant from the Medical Research Council UK to the MRC Clinical Trials Unit [MC_UU_00004/03]. The author has no other competing interests to declare. Sarah L Pett: has received grants from Gilead Sciences, ViiV Healthcare, Janssen-Cilag, Academy of Medical Sciences, EDCTP, NIHR, NIH, and is a member of the TIPAL (Treating people with idiopathic pulmonary fibrosis with the addition of lansoprazole trial, ISRCTN13526307) DSMB. The author has no other competing interests to declare. Graham S Cooke: is a board member of MHRA. The author has no other competing interests to declare. The other authors declare that no competing interests exist.

### Funding

| Funder | Grant reference number | Author |
| --- | --- | --- |
| Medical Research Council | MR/P025064/1 | Graham S Cooke |
| Wellcome Trust | 206/296/Z/17/Z | Graham S Cooke |

The funders had no role in study design, data collection and interpretation, or the decision to submit the work for publication. For the purpose of Open Access, the authors have applied a CC BY public copyright license to any Author Accepted Manuscript version arising from this submission.

### Author contributions

Barnaby Flower, Data curation, Formal analysis, Supervision, Validation, Investigation, Visualization, Writing – original draft, Project administration, Writing – review and editing; Le Manh Hung, Conceptualization, Supervision, Project administration, Writing – review and editing; Leanne Mccabe, Software, Formal analysis, Investigation, Methodology, Writing – original draft, Project administration, Writing – review and editing; M Azim Ansari, Resources, Data curation, Software, Formal analysis, Validation, Investigation, Methodology, Writing – original draft; Chau Le Ngoc, Formal analysis, Validation, Methodology, Project administration; Thu Vo Thi, Hang Vu Thi Kim, Investigation, Project administration, Writing – review and editing; Phuong Nguyen Thi Ngoc, Investigation, Methodology, Project administration; Le Thanh Phuong, Resources, Supervision, Project administration, Writing – review and editing; Vo Minh Quang, Tran Nguyen Bao, Rogier van Doorn, Resources, Supervision, Project administration; Thuan Dang Trong, Supervision, Project administration, Writing – review and editing; Thao Le Thi, Jennifer Ilo Van Nuil, Supervision, Project administration; Cherry Kingsley, Resources, Project administration, Writing – review and editing; David Smith, Resources, Methodology; Richard M Hoglund, Resources, Software, Validation, Investigation, Methodology, Writing – original draft, Writing – review and editing; Joel Tarning, Resources, Formal analysis, Validation, Methodology; Evelyne Kestelyn, Supervision, Validation, Project administration; Sarah L Pett, Supervision, Visualization, Methodology, Writing – review and editing; Hugo Turner, Conceptualization, Supervision, Writing – review and editing; Guy E Thwaites, Conceptualization, Resources, Supervision, Funding acquisition, Project administration, Writing – review and editing; Eleanor Barnes, Conceptualization, Resources, Data curation, Formal analysis, Supervision, Investigation, Visualization, Methodology, Writing – review and editing; Motiur Rahman, Resources, Supervision, Methodology, Project administration; Ann Sarah Walker, Conceptualization, Data curation, Formal analysis, Supervision, Funding acquisition, Validation, Investigation, Visualization, Methodology, Project administration, Writing – review and editing; Jeremy N Day, Conceptualization, Data curation, Formal analysis, Supervision, Validation, Investigation, Methodology, Project administration, Writing – review and editing; Nguyen VV Chau, Resources, Supervision, Funding acquisition; Graham S Cooke, Conceptualization, Data curation, Supervision, Funding acquisition, Validation, Investigation, Visualization, Methodology, Project administration, Writing – review and editing

## Author ORCIDs
Barnaby Flower http://orcid.org/0000-0002-2659-544X
Joel Tarning http://orcid.org/0000-0003-4566-4030
Evelyne Kestelyn http://orcid.org/0000-0002-5728-0918
Guy E Thwaites http://orcid.org/0000-0002-2858-2087
Ann Sarah Walker http://orcid.org/0000-0002-0412-8509
Jeremy N Day http://orcid.org/0000-0002-7843-6280

## Ethics
Patients referred to the trial were initially enrolled into an observational study which included fibroscan assessment and genotyping. Individuals in this cohort found to be potentially eligible for the trial were invited for further screening. All patients provided written informed consent. The trial was approved by the research ethics committees of The Hospital for Tropical Diseases (ref: CS/BND/18/25), Vietnam Ministry of Health (ref: 6172/QĐ-BYTtnam MoH), Imperial College London (ref: 17IC4238), and Oxford University Tropical Research Ethics Committee (ref: 43-17). The study's conduct and reporting is fully compliant with the World Medical Association's Declaration of Helsinki on Ethical Principles for Medical Research Involving Human Subjects. The trial was registered at ISRCTN, registration number is ISRCTN1710027336.

## Decision letter and Author response
Decision letter https://doi.org/10.7554/eLife.81801.sa1
Author response https://doi.org/10.7554/eLife.81801.sa2

---

# Additional files

## Supplementary files
- MDAR checklist
- Source data 1. Pseudo anonymised HCV RNA data for every participant.

## Data availability
The study protocol and processed study data have been uploaded to the ISRCTN registry (ISRCTN17100273; https://doi.org/10.1186/ISRCTN17100273). The data are available under unrestricted access. The raw, pseudo-anonymised viral load data is available in Source Data File 1. The virus sequencing dataset has been uploaded to Dryad (https://datadryad.org) and is available here: https://doi.org/10.5061/dryad.x0k6djhnp. All data generated in this study is provided in the main text, appendix 1 and Source Data File 1.

The following datasets were generated:

| Author(s) | Year | Dataset title | Dataset URL | Database and Identifier |
|---|---|---|---|---|
| Cooke GS, Day JN, Walker AS | 2022 | A pilot study to assess shortened therapy for hepatitis C infected adults in Vietnam | https://doi.org/10.1186/ISRCTN17100273 | ICRCTN, 10.1186/ISRCTN17100273 |
| Flower B, Hung L, McCabe L, Ansari M, Ngoc C, Thi T, Thi Kim H, Thi Ngoc P, Phuong L, Quang V, Trong T, Thi T, Bao T, Kingsley C, Smith D, Hoglund R, Tarning J, Kestelyn E, Pett S, van Doorn R, Nuil J, Thwaites G, Barnes E, Rahman M, Walker A, Day J, Vinh Chau N, Cooke G | 2022 | Ultra-short response-guided Hepatitis C treatment with sofosbuvir and daclatasvir: the SEARCH study HCV sequence data | https://doi.org/10.5061/dryad.x0k6djhnp | Dryad Digital Repository, 10.5061/dryad.x0k6djhnp |

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

## Appendix 1

## Efficacy of ultra-short, response-guided sofosbuvir and daclatasvir therapy for Hepatitis C: a single arm pilot mechanistic study

### Pharmacokinetic and Pharmacodynamic (PK/PD) process

Sofosbuvir and GS-331007 were extracted from 100 µL of plasma using phospholipid removal in the 96-well plate format (Phree, 8E-S133-TGB, Phenomenex), followed by separation on a Gemini, 50 mm × 2.0 mm I.D. 5 µm, column (00B-4435-B0, Phenomenex). Quantification was performed using selected reaction monitoring for the transitions m/z 530.2–>243.2 (sofosbuvir), 536.2–>243.1 (isotope-labelled internal standard for sofosbuvir), 261.3–>113.1 (GS-331007), and 265.3–>113.1 (isotope-labelled internal standard for GS-331007).

The lower limit of quantification (LLOQ) was set to 1.95 ng/mL for sofosbuvir and 20.5 ng/mL for GS-331007. A total of 9 quality control samples (3×low, 3×mid and 3×high concentration) were analysed for each analyte within each batch of clinical samples (96-well plate), resulting in an accuracy of 2.81–2.93% RSE for sofosbuvir and 2.19–3.50% RSE for GS-331007.

DCV was extracted from 100 µL of plasma using supported liquid extraction in the 96-well plate format (ISOLUTE SLE +96 well plate, 820–0200 P01, Biotage), followed by separation on a Gemini, 50 mm × 2.0 mm I.D. 5 µm, column (00B-4435-B0, Phenomenex). Quantification was performed using selected reaction monitoring for the transitions m/z 739.5–>339.3 (DCV) and 747.5–>339.3 (isotope-labelled internal standard for DCV). The LLOQ was set to 1.64 ng/mL for DCV. A total of 9 quality control samples (3×low, 3×mid and 3×high concentration) were analysed within each batch of clinical samples (96-well plate), resulting in an accuracy of 2.46–2.62% RSE for DCV.

### PK/PD analysis

Maximum concentration ($C_{max}$) and time to reach maximum concentration ($t_{max}$) were derived directly from the observed drug concentrations. The software's automatic slope calculator was used to derive the elimination rate ($\lambda$) (adjusted R2 value with uniform weighting) and terminal elimination half-life ($t_{1/2}$). The drug exposure measured as area under the concentration-time curve (AUC) was calculated for each drug using the trapezoidal method. Linear interpolation was used for acceding concentrations and log-linear interpolation for descending concentrations. For the individual analysis, the linear method was used for all measurements due to accumulation between the day 0 and day 28 measurements. Both the AUC to the last time point ($AUC_{last}$, 8 hours for SOF and 24 hours for GS-331007 and DCV) and AUC to infinity ($AUC_{inf}$) were calculated.

A non-compartmental pharmacodynamic analysis were conducted using viral load data from enrolment to day 14 to calculate area under the viral load – time curve and terminal elimination half-life of the viral clearance curve, using the same methodology as explained above. In addition, the relative reduction in viral load between enrolment and day 1, and between enrolment and day 7 were calculated. Ordinary linear regression of drug exposure ($AUC_{last}$) from the individual pharmacokinetic analysis and the outcome measurements were performed in GraphPad Prism 9.3.1 (GraphPad Software, San Diego, California USA).

### Limitations of PK/PD analysis

Two different sampling schedules were used to collect pharmacokinetic samples on day 0 and day 28, resulting in an overlapping sampling profile overall. Therefore, data collected within an individual on a specific day was not dense enough to justify a non-compartment analysis. However, if the data on day 0 and 28 were combined. it resulted in a complete pharmacokinetic profile for the individual. Two separate pharmacokinetic analyses were carried out. A naïve-pooled analysis was performed separately conducted on day 0 and 28 data. This resulted in a summary of exposure (AUC and Cmax) and half-life of the drugs, but it would not be possible to link these measurements to treatment outcome due to the different sampling strategies. Therefore, a second analysis was performed in which the data from day 0 and 28 were pooled for each individual. This resulted in complete pharmacokinetic profiles for each patient, and the pharmacokinetic-pharmacodynamic analysed demonstrated no significant relationship between drug exposure and treatment outcome. Drug accumulation was observed for the sofosbuvir metabolite GS-331007 and DCV between day 0 and 28. However, the individual analysis should still generate median exposure values for each patient, which can be linked with treatment outcome. Another drawback is that no pre-dose samples were collected. Therefore, the non-

compartmental analysis will assume no concentration at day 0 on day 28 for GS-331007 and DCV, even though drug accumulation was observed.

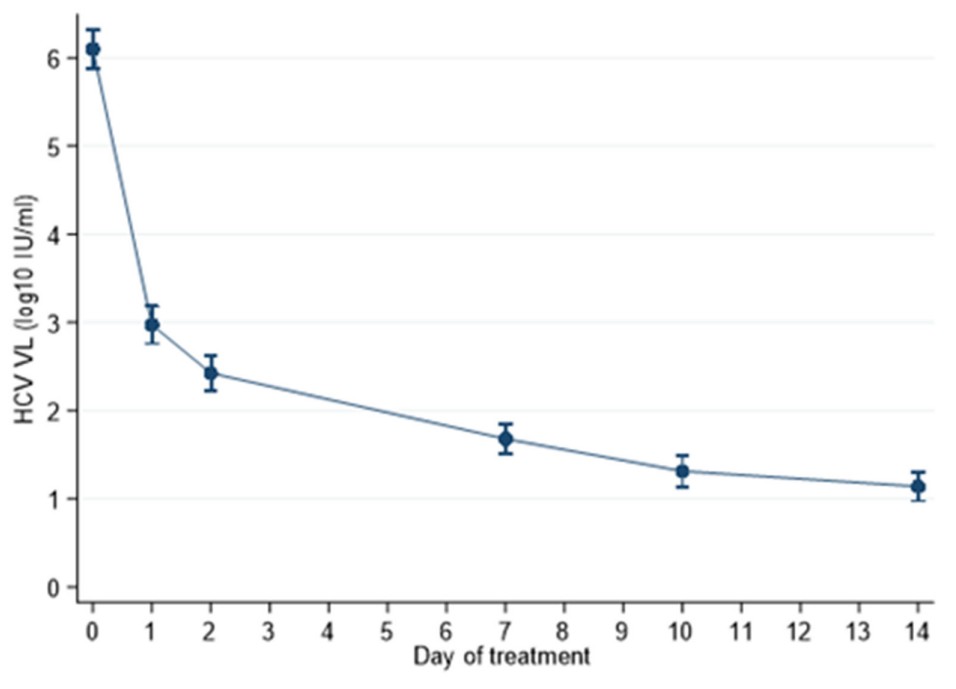

**Appendix 1—figure 1.** Mean (95% CI) HCV RNA by visit day.

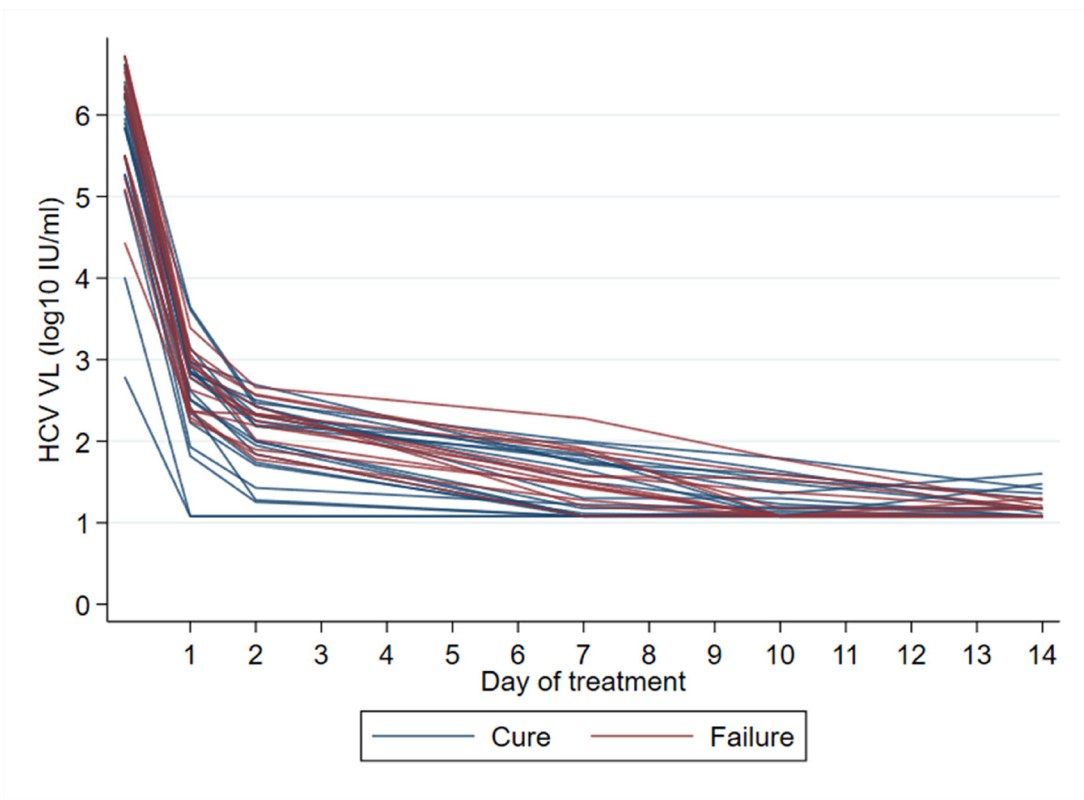

**Appendix 1—figure 2.** HCV RNA kinetics in participants treated with 4 weeks SOF/DCV.

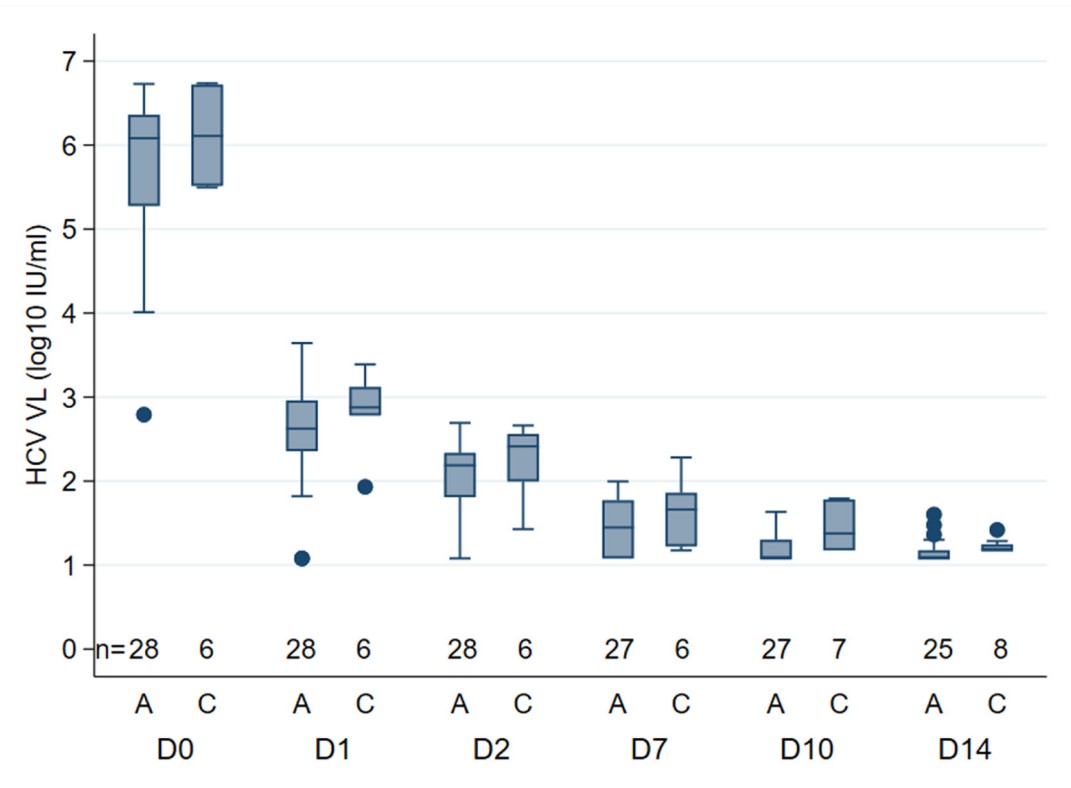

**Appendix 1—figure 3.** Median HCV RNA (log10), by PCR assay, at different time points in participants treated with 4 weeks SOF/DCV A = Abbott Architect (LLOQ = 12 IU/ml). C=COBAS AmpliPrep/COBAS TaqMan HCV Quantitative Test, version 2.0 (Roche Molecular Systems, LLOQ = 15 IU/ml)

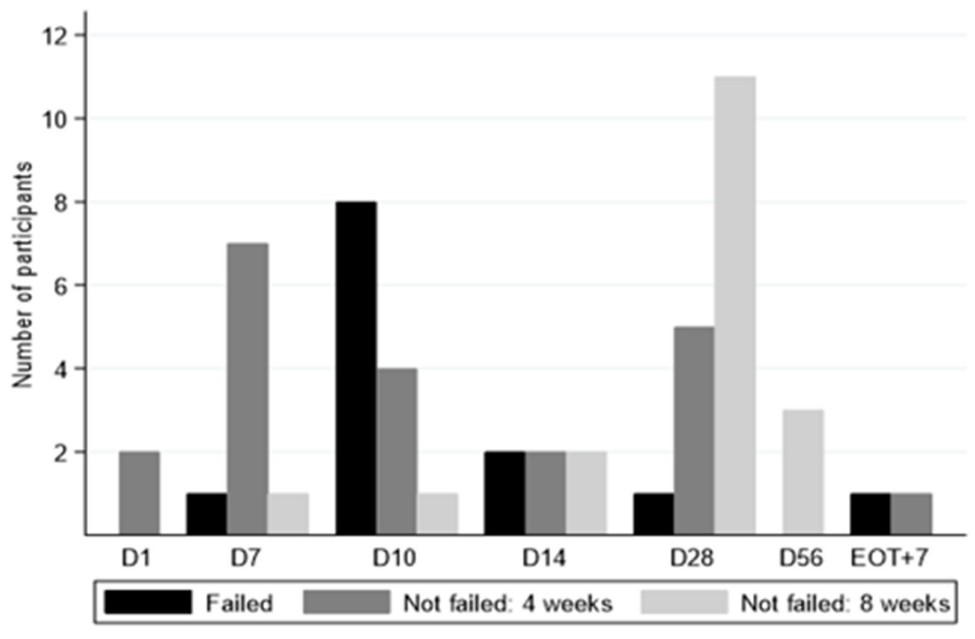

**Appendix 1—figure 4.** Time to viral suppression <LLOQ and eventual treatment outcome. *No treatment failures in 8 week arm. D28 is the EOT visit for those who received 4 weeks. D56 visit is the EOT visit for those who received 8 weeks.

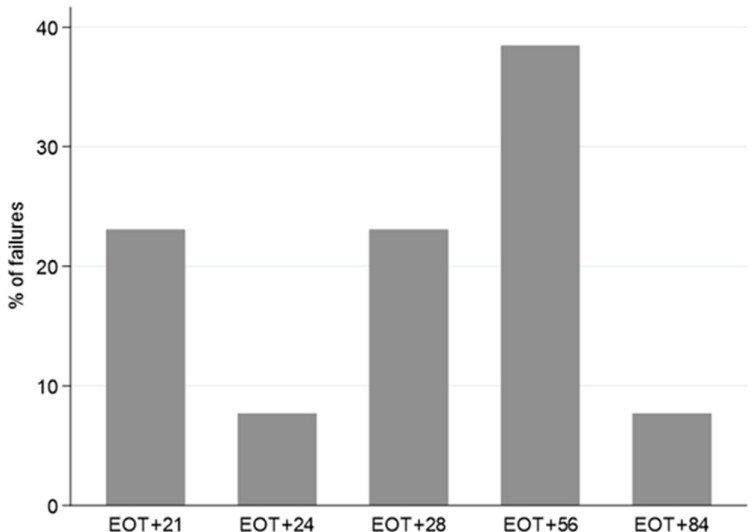

**Appendix 1—figure 5.** Timing of treatment failure (confirmed HCV VL >2000 IU/mL) (n=13). *Note twice weekly sampling in first 4 weeks after EOT, monthly thereafter.

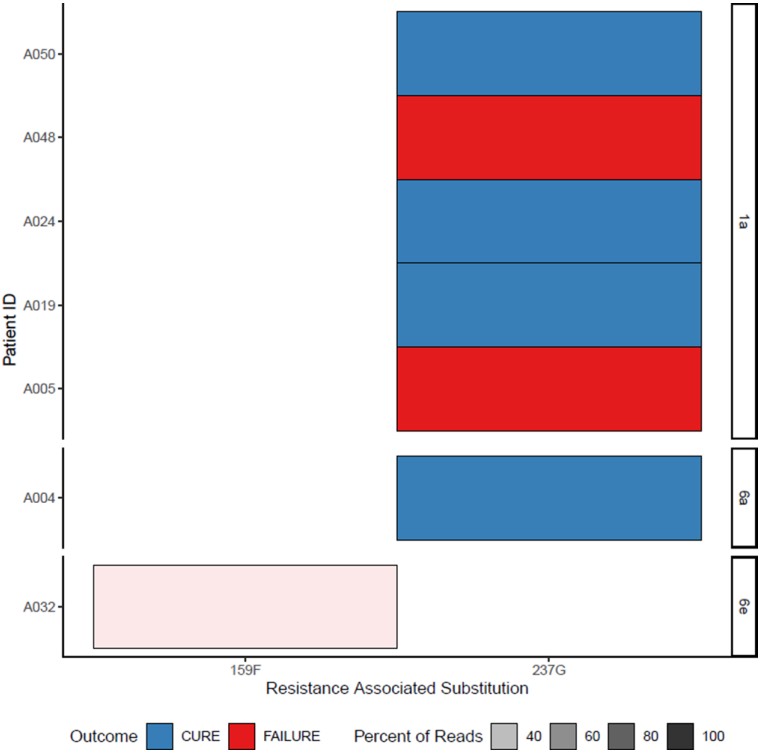

**Appendix 1—figure 6.** All sofosbuvir resistance-associated substitutions at baseline (with treatment outcome).

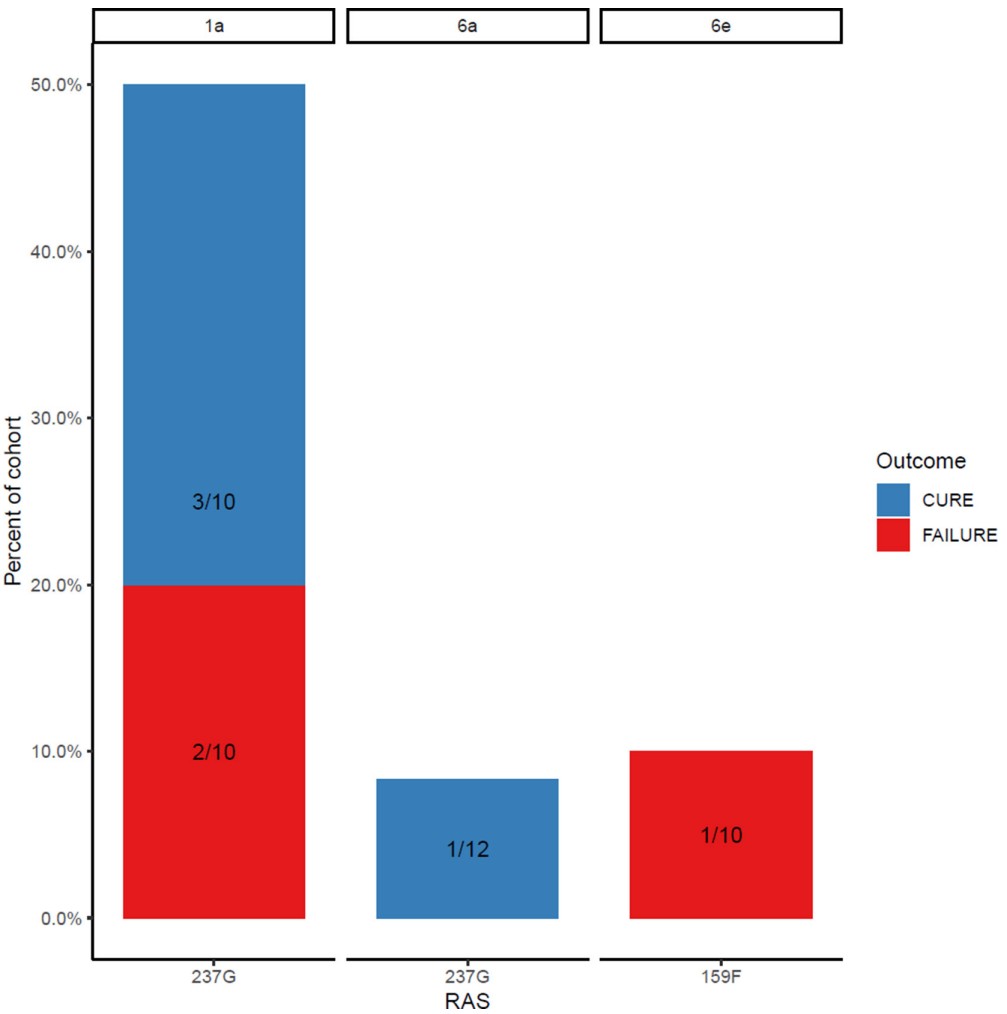

**Appendix 1—figure 7.** Proportion of each subtype with sofosbuvir resistance-associated substitutions at baseline (with treatment outcome).

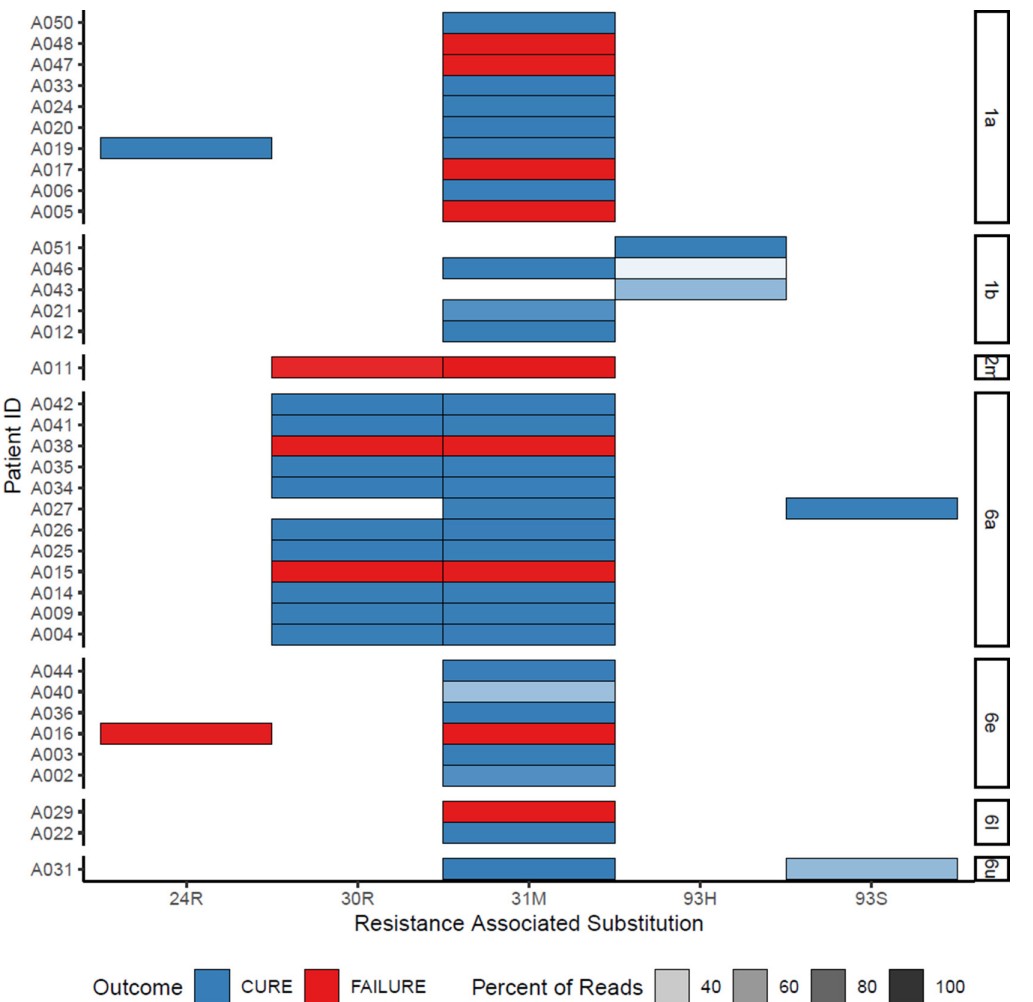

**Appendix 1—figure 8.** All daclatasvir resistance-associated substitutions at baseline (with treatment outcome).

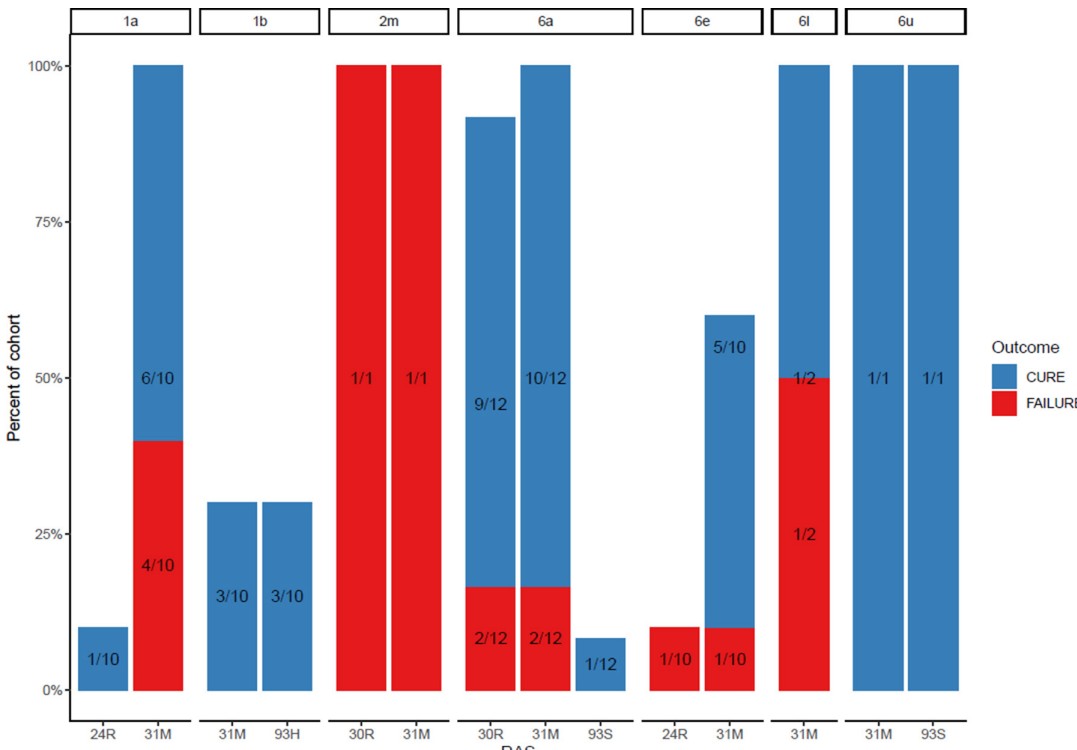

**Appendix 1—figure 9.** Proportion of each subtype with daclatasvir resistance-associated substitutions at baseline (with treatment outcome).

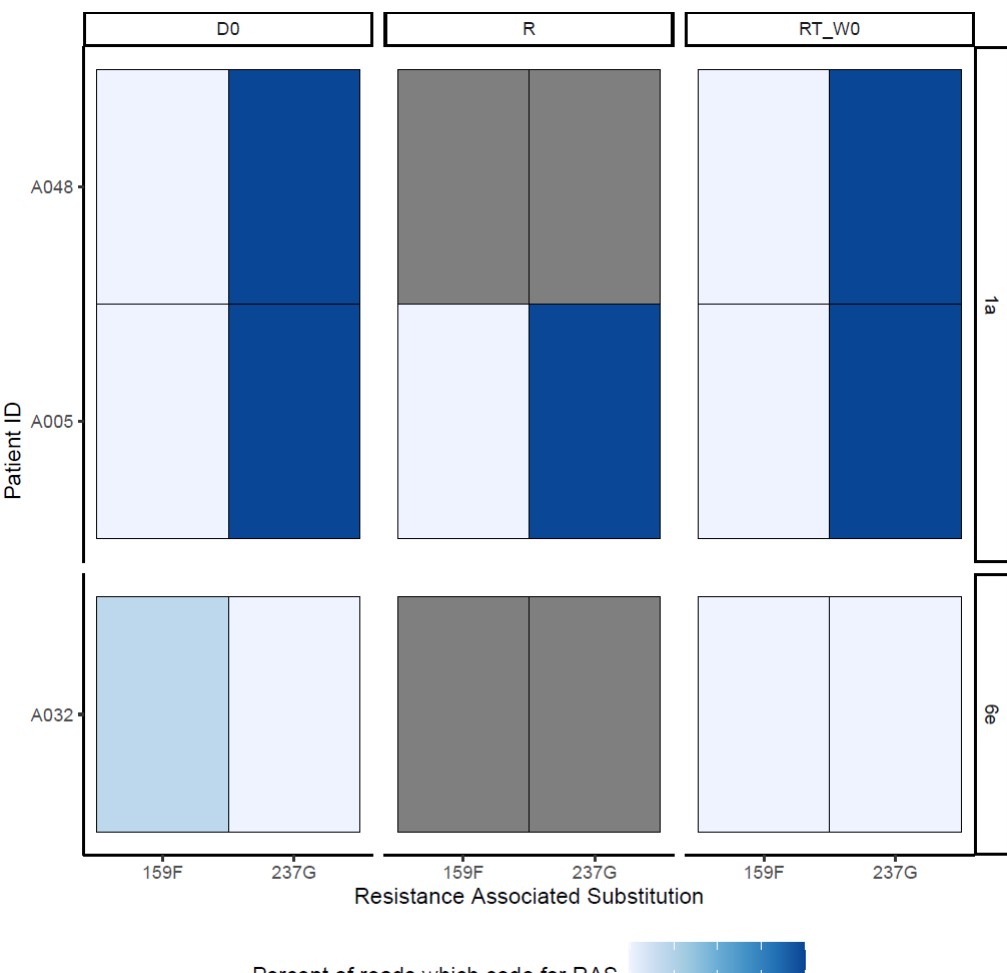

**Appendix 1—figure 10.** SOF-RAS at baseline (**D0**), time of virological rebound (**R**) and start of retreatment (RT_W0) Grey boxes represent missing data.

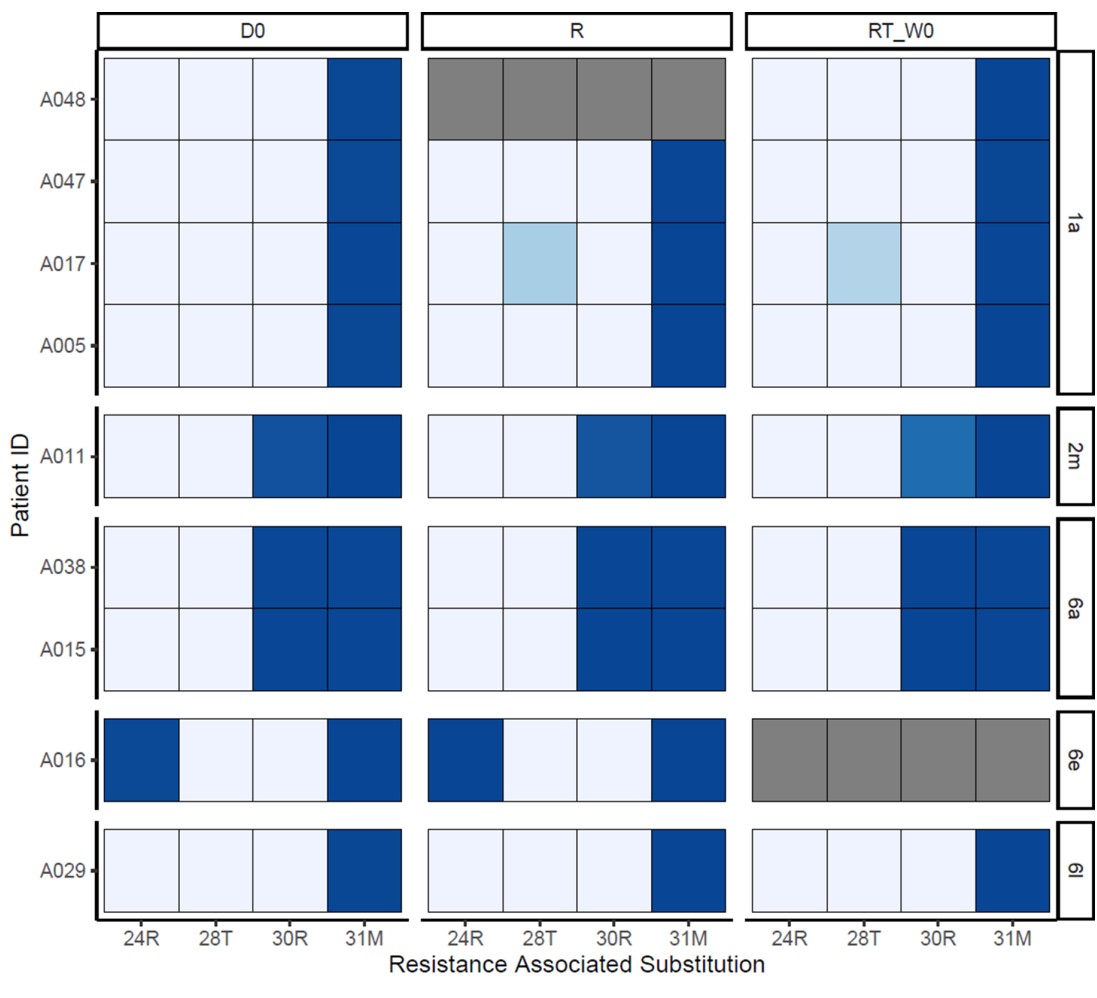

**Appendix 1—figure 11.** DCV-RAS at baseline (**D0**), time of virological rebound (**R**) and start of retreatment (**RT_W0**) *Grey boxes represent missing data.*

**Appendix 1—table 1.** Pharmacokinetic parameters from the naïve-pooled analysis.

| | Sofosbuvir | | GS-331007 | | Daclatasvir | |
|---|---|---|---|---|---|---|
| | **Day 0** | **Day 28** | **Day 0** | **Day 28** | **Day 0** | **Day 28** |
| $C_{max}$ (ng/mL) | 1,320 | 1,070 | 988 | 1,230 | 1,170 | 1,110 |
| $t_{max}$ (h) | 1.00 | 1.00 | 3.00 | 4.00 | 3.00 | 3.00 |
| $t_{1/2}$ (h) | 0.670 | 0.650 | 9.20 | 12.4 | 7.31 | 8.18 |
| $AUC_{last}$ (h×ng/mL)[*] | 1,550 | 1,600 | 10,500 | 14,600 | 11,400 | 12,400 |
| $AUC_{INF}$ (h×ng/mL)[*] | 1,550 | 1,600 | 12,700 | 20,400 | 12,800 | 14,400 |

$C_{max}$ is the maximum observed concentration, $t_{max}$ is the time to reach the maximum concentration, $t_{1/2}$ is the terminal elimination half-life (calculated using the 3–6 last concentration measurements, depending on drug and day), $AUC_{last}$ is the total exposure to the last time point (8 hours for SOF and 24 hours for GS-331007 and DCV), $AUC_{inf}$ is the total exposure extrapolated to infinity.

[*]Extrapolation based on the last observed concentration measurement.

**Appendix 1—table 2.** Pharmacokinetic exposure from the individual analysis and pharmacodynamic parameters.

Pharmacokinetics

|  | Sofosbuvir | GS-331007 | Daclatasvir |
|---|---|---|---|
| $AUC_{last}$ (h×ng/mL) | 1,140 (598-2,150) | 3,430 (2,200-4,720) | 9,770 (5,080-16,200) |
| *Pharmacodynamics* | | | |
| AUC (days ×IU/mL) | 252,000 (19,200-1,370,000) | | |
| $t_{1/2}$ (days) | 2.25 (0.986–5.22) | | |
| %ReductionEnrolment-Day1 | 99.9 (99.0–100) | | |
| %ReductionEnrolment-Day7 | 100 (100–100) | | |

Data is presented as median ($5^{th}$ –$95^{th}$ percentile). $AUC_{last}$ is the total exposure to the last time point (8 hours for SOF and 24 hours for GS-331007 and DCV). $AUC_{14}$ is the area under the viral load-time curve from enrolment (day 0) to day 14, $t_{1/2}$ is the terminal viral half-life (estimated using at least three measurements), %Reduction$_{Enrolment-Day1}$ is the reduction in viral load from enrolment to day 1, %Reduction$_{Enrolment-Day7}$ is the reduction in viral load from enrolment to day 7.

The half-life could not be determined for one participant due to only one sample above the lower limit of quantification.

**Appendix 1—table 3.** Pharmacokinetic-pharmacodynamic linear regression analysis.

| Viral outcome measurement vs. | Sofosbuvir $AUC_{last}$ | | GS-331007 $AUC_{last}$ | | Daclatasvir $AUC_{last}$ | |
|---|---|---|---|---|---|---|
|  | Slope (95% CI) | *p* | Slope (95% CI) | *p* | Slope (95% CI) | p-value |
| Area under the viral load-time curve | −157 (−423–109) | 0.239 | 16.2 (−74.4–107) | 0.719 | −14.2 (−67.1–38.6) | 0.589 |
| Viral elimination half-life | $1.55×10^{-4}$ (−8.70×10$^{-4}$ - 5.60×10$^{-4}$) | 0.662 | $-3.64×10{-5}$ (−2.74×10$^{-4}$ - 2.01×10$^{-4}$) | 0.757 | $2.17×10^{-5}$ (−1.16×10$^{-4}$ - 1.60×10$^{-4}$) | 0.751 |
| Relative reduction in viral load at day 1 | $1.31×10^{-6}$ (−4.54×10$^{-6}$ - 7.16×10$^{-6}$) | 0.652 | $2.67×10{-8}$ (−1.94×10$^{-6}$ - 1.99×10$^{-6}$) | 0.978 | $2.81×10^{-7}$ (−8.62×10$^{-7}$ - 1.42×10$^{-6}$) | 0.621 |
| Relative reduction in viral load at day 7 | $2.53×10^{-7}$ (−2.81×10$^{-7}$ - 7.86×10$^{-7}$) | 0.343 | $5.09×10{-8}$ (−1.29×10$^{-7}$ - 2.31×10$^{-7}$) | 0.569 | $1.44*10^{-8}$ (−9.11×10$^{-8}$ - 1.20×10$^{-7}$) | 0.783 |

95%CI is the 95% confidence interval around the slop.

