## [Editor Report]

Hepatitis C virus (HCV) infection continues to be a global public health problem with over 70 million infected. The current study provides a response to the WHO call for identifying patients with HCV who could be successfully treated with a shorter duration of direct-acting antiviral (DAA) therapy. It provides valuable knowledge to the ongoing research to shorten DAA therapy duration while maintaining high cure rates. Such efforts would impact both treatment access and achieving WHO elimination goals for HCV.

---

## [Decision Letter]

**Decision letter after peer review:**

Thank you for submitting your article "Efficacy of ultrashort, response guided sofosbuvir and daclatasvir therapy for Hepatitis C: a single arm, mechanistic, pilot study" for consideration by *eLife*. Your article has been reviewed by 3 peer reviewers, and the evaluation has been overseen by a Reviewing Editor and Miles Davenport as the Senior Editor. The following individual involved in the review of your submission has agreed to reveal their identity: Harel Dahari (Reviewer #1).

The reviewers have discussed their reviews with one another. Two reviewers were generally positive about the manuscript, while the other reviewer noted that the rationale for an ultrashort therapy to be effective for HCV global elimination is that it does not involve additional testing to identify a subpopulation that may benefit from ultra-short therapy. The Reviewing Editor has drafted this to help you prepare a revised submission.

Essential revisions:

1. The authors cite ref 8 as the main motivation to use RGT based on day 2 (d2) viral load with a cut-off of 500 IU/ml. In ref 8 the length of therapy for those with d2 <500 IU/ml was used to allocate 3 weeks of therapy. It is not clear why in the current work, 4 weeks of therapy was chosen.

2. Some participants seem to have a very low baseline viral load which could indicate a recent HCV infection. Please elaborate.

3. Given the prospective nature of the study and its complexity, there are inherent limitations to the study that is hard to retrospectively fix. As the authors themselves suggest, a protease-inhibitor-free regimen in G1b patients might be incompatible with an efficient enough ultra-short treatment, however, at this stage, this is only a speculation. I would try to compare kinetics to historical cohorts of ultra-short triple (+protease inhibitor) treatment (Lau et al. lancet Gastroenterol Hepatol. 2016)

4. Is there a particular pattern of viral kinetics to 4w cured patients Vs. failures? Figure 1 (Appendix 1) only shows the means of viral load and the general kinetics for the whole population, but individual plots of viral kinetics are not presented although could potentially be useful. Also, according to the presented data, day 7 VL

5. According to Table 3, no significant differences in the host or viral factors were detected between cured or failures of the 4w regimen. However, the low number of patients makes it very difficult to interpret these data and might miss potential differences between these two groups of patients, emphasizing again the difficulty in drawing firm conclusions from this study. In this context, I wonder whether a regression analysis would better define either viral (subtype, RAS) or host factors that are implicated in a 4w duration success.

6. Two assays, Abbott Architect and CobasTaqMan, were noted to be used in the study. Since it was reported that these assays yield different HCV kinetics within the same individual (e.g., DOI: 10.1016/j.jhep.2016.04.006) it would be important to stratify the analyses in the current work also based on these two assays, specifically:

i. In Table 2, four top parameters.

ii. In the viral kinetic analyses (lines 336 – 348).

iii. In Table 3 (days 2, 7, 10, 14 VL).

iv. Provide a figure with viral kinetics based on the two assays (e.g., median with IQR ranges) and perform an analysis to check whether the first phase of decline magnitude from baseline to day 2 and the second phase decline slope (from day 2 until the first time point that the HCV was lower than LLoQ or target not detected, whichever is first) are different as previously discussed, e.g., DOI: 10.1016/j.jhep.2016.05.028.

v. It would be nice to know whether all end of treatment HCV positive cases that went SVR were associated with a particular assay as discussed in DOI: 10.1016/j.jhep.2016.05.028 and/or HCV genotype. This will also clarify the statement in line 346 that begins with "All treatment ….".

7. Reproducibility. I would suggest providing all readers access to the longitudinal raw data (de-identified) that was analyzed for the current study; The processed data uploaded by the authors in ISRCTN does not seem to provide such access. Others may wish to reproduce your analysis and/or try alternative methods to analyze. For example, see the relevant kinetic data-sharing mechanism provided in Supplementary Table 3 in DOI: 10.1111/liv.13335.

8. Limitations. Please consider discussing more study limitations based on the above comments.

[Editors' note: further revisions were suggested prior to acceptance, as described below.]

Thank you for resubmitting your work entitled "Efficacy of ultrashort, response guided sofosbuvir and daclatasvir therapy for Hepatitis C: a single arm, mechanistic, pilot study" for further consideration by *eLife*. Your revised article has been evaluated by Miles Davenport (Senior Editor) and a Reviewing Editor.

The manuscript has been improved but there are some remaining issues that need to be addressed, as outlined below:

1. Please indicate in the viral kinetic file the HCV RNA measurement assay used for each participant.

---

## [Author Response]

Essential revisions:1. The authors cite ref 8 as the main motivation to use RGT based on day 2 (d2) viral load with a cut-off of 500 IU/ml. In ref 8 the length of therapy for those with d2 <500 IU/ml was used to allocate 3 weeks of therapy. It is not clear why in the current work, 4 weeks of therapy was chosen.

The study by Lau et al., which reported 100% cure rate (18/18) used 3 weeks of triple-drug combination treatment SOF + NS5A + π (simeprevir or asunaprevir), and was conducted in a highly selected population of genotype 1b-infected individuals who were mostly females (67%), with a mean age of <40 years, with mild liver disease and low BMI. We used the WHO recommended SOF/DCV (rather than unlicensed triple therapy) as it is the lowest priced and most widely available DAA globally, with the most generic manufacturers. Since we planned to enrol individuals with any genotype 1 or 6 subtype (not just 1b), we opted to use a minimum 4 week treatment duration to provide a greater chance of successful treatment. We have added the following sentence (in red) to the methods section (study design, paragraph 2, page 4).

“Treatment duration was determined using hepatitis C viral load measured 2 days after treatment onset (D2). Participants with viral load <500 IU/ml at D2 (after 2 dose of SOF/DCV) were treated with 4 weeks of SOF/DCV. Those with HCV RNA ≥500 IU/ml received 8 weeks (figure 1). A previous study using this threshold^8^ had found 100% SVR12 following 3 weeks triple therapy (NS5Ai, NS5Bi, PI). We chose a minimum 4-week duration based on our broader inclusion criteria and the use of dual class therapy.”

2. Some participants seem to have a very low baseline viral load which could indicate a recent HCV infection. Please elaborate.

Trial eligibility criteria required that participants had at least one detectable viraemia prior to the screening visit (by quantitative HCV RNA, qualitative assay or HCV genotype), with no intervening undetectable results and plasma HCV RNA >LLOQ at screening. There was no requirement for evidence of chronic infection (>6 months) and therefore there was potential for a patient with acute infection (that might resolve spontaneously) to be enrolled and treated.

Only one patient had a day 0 viral load of <10,000 IU/ml: a 42 year old male with genotype 1b infection who was considered low-risk for acute infection. He had a screening viral load of 274,624 IU/ml, so was eligible for inclusion. However, his day 0 viral load (a few weeks later) was 618 IU/ml, and by 24 hours (day 1) HCV RNA was not detectable. He ultimately cured with 4 weeks therapy. We chose to display range (rather than IQR) in table 1 in part to show this data. To clarify this we have added the following sentence to the results:

“Baseline and clinical characteristics are described in Table 1. One participant, a male with genotype 1b infection who was cured with 4 weeks of therapy, had an HCV viral load of 618 IU/ml on day 0 which may have been consistent with spontaneously resolving acute infection, but could equally reflect fluctuating viraemia. Baseline viral load was >10,000 IU/ml in all other participants, who were all assumed to have chronic infection.”

In addition, we have added the following sentence to ‘Results: Viral kinetics and timing of treatment failure’: “HCV RNA kinetics in those achieving SVR (blue lines) and those not achieving SVR (red lines) with 4 weeks SOF/DCV are shown in appendix 1, figure 2.”

3. Given the prospective nature of the study and its complexity, there are inherent limitations to the study that is hard to retrospectively fix. As the authors themselves suggest, a protease-inhibitor-free regimen in G1b patients might be incompatible with an efficient enough ultra-short treatment, however, at this stage, this is only a speculation. I would try to compare kinetics to historical cohorts of ultra-short triple (+protease inhibitor) treatment (Lau et al. lancet Gastroenterol Hepatol. 2016)

Author response image 1 shows the mean HCV RNA decline on therapy, see also the paper by Lau et al.

**Author response image 1. sa2fig1:** 

In their 2016 study, plasma HCV RNA was measured at baseline (0 h), 1 h, 2 h, 4 h, 8 h, and 24 h after initial DAA dosing, and at days 2, 4, 7, 14, and 21 or at the end of treatment by a COBAS TaqMan 48 analyser (version 2.0 Roche Molecular Systems, Branchburg, NJ, USA, LLOQ of 25 IU/mL and a LLOD of 6 IU/mL). Mean model trajectories were calculated from predicted viral load viral decline in 18 patients treated with three different DAA regimens (6 with SOF+LDV+asunaprevir, 6 with SOF+DCV+asunaprevir, and 6 with SOF+DCV+simeprevir). Their modelled data suggests a broadly similar viral decline to that observed in our study, in which mean HCV RNA was 2.90 (2.75, 3.04) log10 IU/ml at day 1, 2.35 (2.20, 2.50) log10 IU/ml at day 2 and 1.63 (1.50, 1.75) log10 IU/ml at day 7. Lau et al. used a different sampling schedule (7 viral loads in first 48h vs 3 viral loads in first 48h in our study), a different HCV RNA platform (COBAS), and the patients treated with DCV-containing regimens in their study had lower median HCV viral loads than in our study: 5·6 (5·3–6·7) (SOF/DCV/simeprevir) and 5·9 (5·2–6·5) (SOF/DCV/asunaprevir) versus 6.3 (5.8, 6.6) log10 IU/ml in our study. These differences make a detailed comparison of viral kinetics problematic. We have added the following sentence in the discussion.“One important difference was in the treatment regimen, which included a protease inhibitor (simeprevir or asunaprevir). Although NS5A- (DCV) and NS5B- (SOF) inhibitors rapidly eliminate HCV from the blood, second-phase decline in viral load appears to be enhanced by addition of a protease inhibitor^41^. This may be crucial in sustaining high rates of cure with shortened therapy. Viral kinetics in our participants were broadly similar to those observed in patients treated with DCV-containing regimens in the study by Lau et al., with a rapid first phase viral decline leading to an approximate 4 log10 IU/ml decline in HCV RNA in the first 48h of treatment. However, a detailed comparison of viral kinetics is limited by differences in sampling schedule, baseline viral loads and the HCV PCR platforms used.”

4. Is there a particular pattern of viral kinetics to 4w cured patients Vs. failures? Figure 1 (Appendix 1) only shows the means of viral load and the general kinetics for the whole population, but individual plots of viral kinetics are not presented although could potentially be useful. Also, according to the presented data, day 7 VL

We have added appendix 1 – figure 2 shown above. We have added the following text to ‘Results: Viral kinetics and timing of treatment failure’: “HCV RNA kinetics in all participants treated with 4 weeks SOF/DCV is shown in appendix 1, figure 2, with cures (blue lines) distinguished from those experiencing treatment failure (red lines). Even though the numbers are small, this helps illustrate that early on-treatment response alone may be of limited value in determining cure with ultra-short therapy.”

5. According to Table 3, no significant differences in the host or viral factors were detected between cured or failures of the 4w regimen. However, the low number of patients makes it very difficult to interpret these data and might miss potential differences between these two groups of patients, emphasizing again the difficulty in drawing firm conclusions from this study. In this context, I wonder whether a regression analysis would better define either viral (subtype, RAS) or host factors that are implicated in a 4w duration success.

We thank the reviewers for this suggestion. We previously avoided doing regression analysis due to the high number of host variables (age, gender, BMI, liver function tests) and virus variables (baseline viral load, viral kinetics, subtype, RAS) variables in a small sample size. We have performed regression analysis in response to this review and the only statistically significant result was that there was a significant *reduction* in treatment failure in participants with 2 resistance mutations compared to those with 1 (p=0.01). This is counter-intuitive and is likely a reflection of the small numbers involved: (6 (5 failures) with 1 RAS and 21 (8 failures) with 2 RAS). We would prefer not to present a regression analysis because this was not pre-defined in the study protocol. The study was not powered to detect differences between 4 week cures and 4 week treatment failures and we feel this post-hoc analysis may confuse things.

6. Two assays, Abbott Architect and CobasTaqMan, were noted to be used in the study. Since it was reported that these assays yield different HCV kinetics within the same individual (e.g., DOI: 10.1016/j.jhep.2016.04.006) it would be important to stratify the analyses in the current work also based on these two assays, specifically:i. In Table 2, four top parameters.

Added.

ii. In the viral kinetic analyses (lines 336 – 348).

Added*:*

“Since the two HCV assays used in our study have previously been shown to yield different HCV RNA results in the same individuals on therapy^36^, we conducted additional analysis of viral kinetics stratified by platform. We found no evidence of a difference between platforms in terms of proportion of participants with undetectable viral load at different timepoints (table 2, table 3), or in terms of first phase (day 0 to day 2) or second phase (day 2 to first HCV RNA < LLOQ) viral decline on treatment. However numbers were small.”

iii. In Table 3 (days 2, 7, 10, 14 VL).iv. Provide a figure with viral kinetics based on the two assays (e.g., median with IQR ranges) and perform an analysis to check whether the first phase of decline magnitude from baseline to day 2 and the second phase decline slope (from day 2 until the first time point that the HCV was lower than LLoQ or target not detected, whichever is first) are different as previously discussed, e.g., DOI: 10.1016/j.jhep.2016.05.028.

We have added appendix 1 – figure 3: Boxplot showing median HCV RNA decline, by assay, at different timepoints in the 34 participants who had a rapid virological response. We found no evidence of a difference in viral decline between the two assays.

v. It would be nice to know whether all end of treatment HCV positive cases that went SVR were associated with a particular assay as discussed in DOI: 10.1016/j.jhep.2016.05.028 and/or HCV genotype. This will also clarify the statement in line 346 that begins with "All treatment ….".

Three patients who achieved SVR had detectable HCV RNA at EOT, of which two were on Abbott and one on COBAS. These are insufficient numbers to draw conclusions about the assays so we have not commented on this in the manuscript.

7. Reproducibility. I would suggest providing all readers access to the longitudinal raw data (de-identified) that was analyzed for the current study; The processed data uploaded by the authors in ISRCTN does not seem to provide such access. Others may wish to reproduce your analysis and/or try alternative methods to analyze. For example, see the relevant kinetic data-sharing mechanism provided in Supplementary Table 3 in DOI: 10.1111/liv.13335.8. Limitations. Please consider discussing more study limitations based on the above comments.

In Discussion we have added:

“Another potential limitation relates to our use of two different HCV RNA platforms which have previously been shown to give discrepant results in the same individuals^56^. The Abbott Architect has a lower LLOQ than the COBAS AmpliPrep/COBAS TaqMan and may detect HCV RNA for longer on therapy than the COBAS^36^. We found no evidence of difference in viral decline by platform but it would have been preferable to use one platform to determine day 2 HCV RNA in all participants.”

[Editors' note: further revisions were suggested prior to acceptance, as described below.]

The manuscript has been improved but there are some remaining issues that need to be addressed, as outlined below:1. Please indicate in the viral kinetic file the HCV RNA measurement assay used for each participant.

I have now indicated which platform each HCV VL was performed on in the attached file named “Source Data File 1 Pseudo anonymised HCV RNA data_with platforms_13DEC22”